# Fast Direct: Query-Efficient Online Black-box Guidance for Diffusion-model Target Generation

**Kim Yong Tan** [1]   **Yueming Lyu** [2,3*]  **Ivor Tsang**[1,2,3]   **Yew-Soon Ong**[1,2,3]

[1]College of Computing and Data Science, Nanyang Technological University, Singapore
[2]Centre for Frontier AI Research, Agency for Science, Technology and Research, Singapore
[3]Institute of High Performance Computing, Agency for Science, Technology and Research, Singapore

{KIMYONG001,ASYSOng}@ntu.edu.sg
{Lyu_Yueming,Ivor_Tsang}@cfar.a-star.edu.sg

## Abstract

Guided diffusion-model generation is a promising direction for customizing the generation process of a pre-trained diffusion model to address specific downstream tasks. Existing guided diffusion models either rely on training the guidance model with pre-collected datasets or require the objective functions to be differentiable. However, for most real-world tasks, offline datasets are often unavailable, and their objective functions are often not differentiable, such as image generation with human preferences, molecular generation for drug discovery, and material design. Thus, we need an **online** algorithm capable of collecting data during runtime and supporting a **black-box** objective function. Moreover, the **query efficiency** of the algorithm is also critical because the objective evaluation of the query is often expensive in real-world scenarios. In this work, we propose a novel and simple algorithm, **Fast Direct**, for query-efficient online black-box target generation. Our Fast Direct builds a pseudo-target on the data manifold to update the noise sequence of the diffusion model with a universal direction, which is promising to perform query-efficient guided generation. Extensive experiments on twelve high-resolution ($1024 \times 1024$) image target generation tasks and six 3D-molecule target generation tasks show **6$\times$** up to **10$\times$** query efficiency improvement and **11$\times$** up to **44$\times$** query efficiency improvement, respectively. Our implementation is publicly available at: https://github.com/kimyong95/guide-stable-diffusion/tree/fast-direct

## 1 Introduction

Diffusion models have become the state-of-the-art generative model for image synthesis (Ho et al., 2020; Nichol & Dhariwal, 2021; Dhariwal & Nichol, 2021) and video synthesis (Ho et al., 2022), etc. Its remarkable success is due to its powerful capability in modeling complex multi-mode high-dimensional data distributions.

One promising direction for utilizing the generative power of diffusion models is through target generation, which allows users to customize the generation process to meet specific downstream objectives, effectively extending the models' capabilities beyond synthesis problems to engineering optimization and science discovery problems, such as image generation with human preferences, drug discovery (Corso et al., 2022; Guan et al., 2023) and material design (Vlassis & Sun, 2023; Giannone et al., 2023).

The pre-trained diffusion models often struggle to generate desired samples for these applications, especially when the target data lies outside the training data distribution. Therefore, the target generation often involves model fine-tuning or guidance techniques. Krishnamoorthy et al. (2023) proposes to train the diffusion model with re-weighted training loss, while Clark et al. (2023); Black et al. (2023); Fan et al. (2024); Yang et al. (2024) advocates for fine-tuning the parameters of the pre-

---

*Corresponding author

trained model. As opposed to training-time approaches, Bansal et al. (2023) introduces an inference-time approach, which replaces the classifier in classifier guidance (Dhariwal & Nichol, 2021) with a differentiable objective function to achieve the downstream target. However, the requirement of the differentiable objective limits its practical usage for real applications with black-box feedback.

Most real-world applications of target generation involve evaluating **black-box objectives** in an **online** manner. For example, image generation with human preferences requires human users to evaluate the generated images; drug design requires real-world experiments to evaluate the generated molecules. These applications typically require several iterative query-feedback cycles with a black-box objective to achieve satisfactory target generation. As objective evaluations are often expensive, it becomes crucial to develop **query-efficient** algorithms to minimize the cost of these evaluations.

Although the **online black-box** target generation tasks are important, the existing works are not suitable to address this task. Bansal et al. (2023); Krishnamoorthy et al. (2023); Lu et al. (2023) require training an offline guidance model with pre-collected data, while Bansal et al. (2023); Clark et al. (2023); Prabhudesai et al. (2023); He et al. (2023) require the objective function to be differentiable. Most recently, Black et al. (2023); Fan et al. (2024); Yang et al. (2024) can be employed for online black-box target generation, but they require online updates of the huge number of parameters, which is both time-consuming and not query-efficient.

In Section 3.1, we first propose a novel guided noise sequence optimization (GNSO) technique to guide the diffusion model sampling process towards a given target. GNSO updates the noise sequence in a *universal direction* on the data manifold. Our GNSO is **efficient**: it enables fast adaptation (with around only 50 steps) from any initial generation towards the given input target. Moreover, GNSO is **robust**: empirically, even if the input target image is noisy, it can still generate a clear image semantically similar to the target image. The GNSO itself cannot directly handle black-box target generation tasks, but it provides a backbone for further designing black-box algorithms.

In Section 3.2, based on our GNSO technique, we further propose a novel algorithm, **Fast Direct**, to tackle the **online black-box** target generation tasks in a **query-efficient** manner. Fast Direct builds a pseudo-target on the data manifold as the input for our GNSO, to guide the diffusion model at inference time. Notably, our Fast Direct is easy to implement and supports any stochastic diffusion scheduler. Moreover, our Fast Direct provides a very flexible framework for extension. Any designs of update methods for the pseudo-target can serve as a plug-in for our Fast Direct.

In Section 4, we evaluate our Fast Direct algorithm on the real-world applications: high-resolution ($1024 \times 1024$) image target generation tasks and 3D-molecule target generation tasks. For image tasks, we employ the black-box API of the modern Large Language Model (LLM), Gemini 1.5, as the black-box objective, which is much more practical compared with the synthetic toy score function used in the literature. Extensive experiments on twelve image target generation tasks and six 3D-molecule target generation tasks show significant query efficiency improvement: **6**-times up to **10**-times query efficiency improvement on image tasks and **11**-times up to **44**-times query efficiency improvement on 3D-molecule tasks, compared with baselines. Additionally, we evaluate Fast Direct on compressibility, incompressibility, and atheistic quality tasks, demonstrating its generalization ability on the unseen prompts. Our contributions are summarized as follows:

- We propose a novel guided noise sequence optimization (GNSO) technique to guide the diffusion model sampling process toward a given target in an efficient and robust manner.
- Based on GNSO, we further propose a novel algorithm, Fast Direct, to address online black-box target generation tasks in a query-efficient manner.
- Fast Direct achieves significant query efficiency improvement in real-world applications: high-resolution image target generation tasks and 3D-molecule target generation tasks. Additionally, we demonstrate its generalization capability for unseen prompts.

## 2 RELATED WORKS

### 2.1 GUIDED GENERATION

Diffusion guided generation (also called inference-time guidance) refers to the technique to guide the sampling trajectory of the pre-trained diffusion to generate target data (Croitoru et al., 2023;

Chen et al., 2024). The key advantage of this approach is that it does not require updating the model parameters, which is computationally expensive, especially for large models.

Dhariwal & Nichol (2021) proposed classifier guidance. However, it requires a guidance model trained on noisy images with different noise scales, which is generally not readily available and often requires training from scratch for each domain. Chung et al. (2022); Bansal et al. (2023); He et al. (2023) extend the classifier guidance by using a differentiable objective function defined only for clean data. Instead of using noisy images, the predicted clean images at each sampling step are used as the input for the guidance objective function. In this way, the guidance process can operate on the clean image space. While the predicted clean image is naturally imperfect, empirically it still provides informative feedback to guide image generation (Bansal et al., 2023).

However, this approximation can harm image quality. Bansal et al. (2023) proposed universal guidance that comprises backward guidance followed by a self-recurrence step to preserve image quality. On the other hand, He et al. (2023) leverages the differentiability of a well-trained auto-encoder to project the image to the data manifold and thus preserve image quality during the guidance process. Instead of guiding the sampling trajectory, Karunratanakul et al. (2024); Eyring et al. (2024) treat the diffusion process as a black-box and only optimize the initial (prior) noise.

While the aforementioned approaches assume a differentiable objective function, Lu et al. (2023) tackles black-box objective $f$ by learning a differentiable proxy neural network $h$ to match their gradients, i.e., $\nabla f \approx \nabla h$. Li et al. (2024) eliminates the need for a differentiable proxy model by employing importance sampling weighted by the objective values during the sampling process. Most recently, DNO (Tang et al., 2024) proposes optimizing the diffusion noise sequence by using ZO-SGD (Nesterov & Spokoiny, 2017) to tackle the black-box objective function. However, it runs at the instance level; namely, each run only produces one image.

## 2.2 DIFFUSION-MODEL FINE-TUNING

Diffusion-model fine-tuning refers to the technique that updating the pre-trained diffusion-model parameters to improve its performance on a specific use case. In this section, we review the fine-tuning methods that support online learning of the black-box objective function.

Black et al. (2023); Fan et al. (2024) formulate the diffusion fine-tuning problem as a reinforcement learning (RL) problem within Markov Decision Processes (MDPs), and propose an iterative algorithm to fine-tune the diffusion model by using Proximal Policy Optimization (PPO) (Schulman et al., 2017; Uehara et al., 2024) loss function. Fan et al. (2024) integrates the PPO with a KL regularization to prevent the fine-tuned model from deviating too much from the pre-trained model.

On the other hand, Yang et al. (2024) does not require an absolute objective value; instead, it uses the relative reward on a pair of samples by integrating DPO (Direct Preference Optimization) (Rafailov et al., 2024), a technique for fine-tuning large language models, into the diffusion model.

Fine-tuning the diffusion model requires a large amount of GPU memory. Existing work mitigates memory consumption by using the LoRA technique (Low-Rank Adaptation) (Hu et al., 2021) and only fine-tunes parameters of the attention blocks in UNet. We categorize the related works based on whether they support the online and black-box objective tasks in Appendix F Table 2.

## 3 METHODS

In this section, we first present a novel inference-time guidance generation method by guided noise sequence optimization. Based on this method, we further present our query-efficient online black-box guidance algorithm, Fast Direct, to address online black-box guidance tasks.

## 3.1 NOISE SEQUENCE OPTIMIZATION WITH TARGET GUIDANCE

Take i.i.d. Gaussian samples $\{\boldsymbol{\epsilon}_0, \cdots, \boldsymbol{\epsilon}_K\} \sim \mathcal{N}(\boldsymbol{0}, \boldsymbol{I})$, for $k \in \{1, \cdots, K\}$, the inference process of the diffusion model can be formulated as follows:

$$\boldsymbol{x}_k = \mathcal{S}_\theta(\boldsymbol{x}_{k-1}, \boldsymbol{\epsilon}_k), \tag{1}$$

---

**Algorithm 1:** Guided Noise Sequence Optimization

---

**Input:** Stepsize $\alpha$, target data $\boldsymbol{x}^*$, number of diffusion sampling steps $K$, pre-trained diffusion sampler $\mathcal{S}_\theta$, number of iterations $T$.

**Output:** Generated target $\boldsymbol{x}_K$.

1   Take i.i.d. noise $\{\boldsymbol{\epsilon}_0, \cdots, \boldsymbol{\epsilon}_K\} \sim \mathcal{N}(\boldsymbol{0}, \boldsymbol{I})$

2   Compute the norm of noise $\epsilon_k^{\text{norm}} = \|\boldsymbol{\epsilon}_k\|$ for $k \in \{0, \cdots, K\}$.

3   **for** $t \leftarrow 1$ **to** $T$ **do**

4      $\boldsymbol{x}_0 \leftarrow \boldsymbol{\epsilon}_0$

5      **for** $k \leftarrow 1$ **to** $K$ **do**

6         $\boldsymbol{x}_k \leftarrow \mathcal{S}_\theta(\boldsymbol{x}_{k-1}, \boldsymbol{\epsilon}_k)$

7      **for** $k \leftarrow 0$ **to** $K$ **do**

8         $\hat{\boldsymbol{\epsilon}}_k \leftarrow \boldsymbol{\epsilon}_k + \alpha(\boldsymbol{x}^* - \boldsymbol{x}_K)$

9         $\boldsymbol{\epsilon}_k \leftarrow \frac{\hat{\boldsymbol{\epsilon}}_k}{\|\hat{\boldsymbol{\epsilon}}_k\|} \epsilon_k^{\text{norm}}$

---

Line 4 - Line 6: Call diffusion-model sampler to generate data $\boldsymbol{x}_K$.

Line 7 - Line 9: Update the noise $\boldsymbol{\epsilon}_k$ by moving along direction $\boldsymbol{x}^* - \boldsymbol{x}_K$.

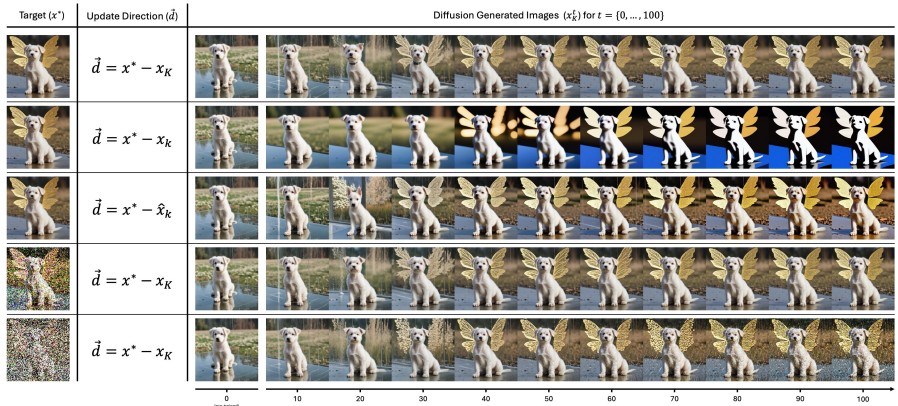

Figure 1: Demonstration of guided generation for a given target by Algorithm 1. Column 2 (Update Direction) indicates the update term of Algorithm 1, Line 8. Rows 1 to 3 analyze how different update directions can affect the generated images. Rows 4 and 5 show that by using the update direction of $\boldsymbol{x}^* - \boldsymbol{x}_K$, the diffusion model can generate visually satisfying images even when the target image is noisy. The noisy target image ($\boldsymbol{x}^*$) of the row 4 is obtained by clean image added with noise $\mathcal{N}(\boldsymbol{0}, \boldsymbol{I})$, and row 5 is added with noise $\mathcal{N}(\boldsymbol{0}, 9 \times \boldsymbol{I})$.

where $\mathcal{S}_\theta(\cdot, \cdot)$ denotes the diffusion model sampler that depends on the concrete SDE solver used, $\boldsymbol{x}_0 = \boldsymbol{\epsilon}_0$ is a Gaussian sample, and $\boldsymbol{x}_K$ denotes the generated data.

*Question 1: Given an input target $\boldsymbol{x}^*$, can we guide a pre-trained diffusion model at inference time to generate the target data efficiently and robustly?*

To answer this question, we need to develop an algorithm to address two challenges simultaneously: (1) Guidance Efficiency: the algorithm can generate the target data $\boldsymbol{x}^*$ by using only a few guidance update steps. (2) Robust to noisy target: given a noisy target (e.g., a noise-perturbed image), the algorithm can generate meaningful target data (e.g., a clear image).

We propose a novel guided noise sequence optimization method to address the above two challenges, which is presented in Algorithm 1. Our method optimizes the noise sequence by updating the noise along the direction on the data manifold. Specifically, Algorithm 1 consists of two procedures: a data generation procedure and a noise update procedure. In the data generation procedure, Line 4 - Line 6 in Algorithm 1 employ current noise sequence $\{\boldsymbol{\epsilon}_0, \cdots, \boldsymbol{\epsilon}_K\}$ to generate a data $\boldsymbol{x}_K$. In the noise update procedure, Lines 7 - 9 in Algorithm 1, we update each noise $\boldsymbol{\epsilon}_k$ by moving along a universal direction $\boldsymbol{x}^* - \boldsymbol{x}_K$. Then, Algorithm 1 continues to employ the updated noise to generate a new data $\boldsymbol{x}_K$. By looping the data generation procedure and noise update procedure, Algorithm 1 can generate a clear and meaningful target.

The direction $x^* - x_K$ point from the current generation data $x_K$ to the target $x^*$. Intuitively, moving in this direction will shift the diffusion sampling trajectory toward the target. We highlight the advantage of updating noise with the *universal direction* $x^* - x_K$ compared with direction $x^* - x_k$ as the following remark.

**Remark:** In our Algorithm 1, we guided the noise update by moving along a universal direction $x^* - x_K$. This direction lies in the data manifold, which alleviates the quality degeneration of the generated data. We employ the direction $x^* - x_K$ instead of the direction $x^* - x_k$ because the internal $x_k$ during inference sampling process may be far away from the data manifold (e.g., the $x_k$ can be a very noisy and distorted image), which may degenerate the quality.

We empirically evaluate Algorithm 1 with different update directions: $\vec{d} = x^* - x_K$ and $\vec{d} = x^* - x_k$. Additionally, we consider the approximation $x_K \approx \hat{x}_k$[1]. Lastly, we set the noisy target to test the robustness. We use the stable-diffusion model as the pre-trained model. The stepsize $\alpha$ is set to $\alpha = 2 \times 10^{-3}$. We present the generated images every 10 iterations for each case in Figure 1.

From Figure 1, we can observe that Algorithm 1 (with update direction $x^* - x_K$) can successfully guide the diffusion process towards the target image, while $x^* - x_k$ leads to a degenerated image. The approximation $x^* - \hat{x}_K$ can achieve comparable target image but not as accurate as $x^* - x_K$.

Interestingly, even the input target $x^*$ is perturbed by noise and lies out of the data manifold, Algorithm 1 will generate a pseudo-data on data manifold that is similar to the noisy target $x^*$ but with much higher quality compared with the input noisy target. This property is important for designing black-box guidance methods because the update direction is an approximation or estimation of the true gradient of the black-box objective.

Moreover, we present the updating direction $x^* - x_{K'}$ for $K' \in \{K, K/2, K/4, K/8\}$ in Appendix C. We observe that the generated image quality decreases as the $x_{K'}$ becomes noisier.

## 3.2 QUERY-EFFICIENT ONLINE BLACK-BOX GUIDANCE

In section 3.1, we address the diffusion guidance sampling with a given optimal target input. However, in reality, we often only have access to the black-box objective function $f(x)$, where the optimal target data $x^* = \arg\min f(x)$ is unknown. In this section, we further investigate the diffusion guidance sampling with only black-box objective feedback. For this task, it is natural to ask the following question.

*Question 2: Can we guide a pre-trained diffusion model at inference time to generate target data with only black-box objective feedback in a query-efficient online manner?*

To answer this question, we need to address two challenges: (1) the Black-box challenge and (2) the Online guidance challenge. The black-box challenge means that we cannot access the gradient of the objective. The online guidance challenge means we don't have a prior dataset to train a surrogate (classifier) for guidance; we can only access the black-box objective through query feedback online. Usually, the query evaluation is expensive. Thus, query efficiency is critical.

We start addressing the above two challenges based on our target guidance generation Algorithm 1. Although Algorithm 1 itself cannot handle black-box guidance tasks because it needs input a target $x^*$, it provides a basis for us to design query-efficient online black-box guidance methods. To be specific, Algorithm 1 enables us to generate a target even updating with a noisy direction $x^* - x_K$. This property is important for black-box guidance tasks because we can use a noisy estimation of the gradient of the black-box objective to guide the generation.

The key idea is to set a pseudo target $\hat{x}^*$ to guide the generation process based on our Algorithm 1. We present our Fast Direct method as in Algorithm 2. In Algorithm 2, we call Algorithm 1 inside the for-loop w.r.t. the number of batch queries $t$. The only difference compared with Alg. 1 is that we set a pseudo target $\hat{x}^*$ in Line 10 of Algorithm 2 instead of a given fixed target $x^*$.
The choice of models for updating the pseudo target $\hat{x}^*$ in Algorithm 2 is flexible, which supports various black-box target generation method designs based on our Fast Direct algorithm framework. In this work, we set the pseudo target $\hat{x}^*$ through nonparametric methods without additional training.

---

[1] The $\hat{x}_k$ is the "predicted clean image" at step $k$, see the Eq.12 of DDIM (Song et al., 2020). Note that it is undefined for the initial step, so we simply set $\hat{x}_0 \coloneqq \hat{x}_1$.

---

**Algorithm 2:** Fast Direct

---

**Input:** Max number of batch queries $N$, batch size $B$, step-size $\alpha$, number of diffusion sampling steps $K$, pre-trained diffusion sampler $\mathcal{S}_\theta$.

**Output:** Set of optimized samples $\{x_K^1, \cdots, x_K^B\}$, pseudo target model that learns from dataset $\mathcal{D}$.

1 Initialize $\mathcal{D} \leftarrow \{\}$
2 **for** $i \leftarrow 1$ **to** $N$ **do**
3    **do parallel for** $B$ **instances**
4       Take i.i.d. noise $\{\epsilon_0, \cdots, \epsilon_K\} \sim \mathcal{N}(\mathbf{0}, \mathbf{I})$.
5       Compute the norm of noise $\epsilon_k^{\text{norm}} = \|\epsilon_k\|$ for $k \in \{0, \cdots, K\}$.
6       **for** $t \leftarrow 1$ **to** $i$ **do**
7          $\boldsymbol{x}_0 \leftarrow \boldsymbol{\epsilon}_0$
8          **for** $k \leftarrow 1$ **to** $K$ **do**
9             $\boldsymbol{x}_k \leftarrow \mathcal{S}_\theta(\boldsymbol{x}_{k-1}, \boldsymbol{\epsilon}_k)$
10       Set a pseudo target $\hat{\boldsymbol{x}}^*$ by the pseudo target model with input $\boldsymbol{x}_K$.
11       **for** $k \leftarrow 0$ **to** $K$ **do**
12          $\hat{\boldsymbol{\epsilon}}_k \leftarrow \boldsymbol{\epsilon}_k + \alpha(\hat{\boldsymbol{x}}^* - \boldsymbol{x}_K)$
13          $\boldsymbol{\epsilon}_k \leftarrow \frac{\hat{\boldsymbol{\epsilon}}_k}{\|\hat{\boldsymbol{\epsilon}}_k\|} \epsilon_k^{\text{norm}}$
14       Query black-box objective score $y \leftarrow f(\boldsymbol{x}_K)$.
15       Increment dataset $\mathcal{D} \leftarrow \mathcal{D} \cup \{(\boldsymbol{x}_K, y)\}$.
16    Update pseudo target model with dataset $\mathcal{D}$.

---

Specifically, we employ two methods for setting the pseudo target $\hat{\boldsymbol{x}}^*$, namely, Gaussian process (GP) update and historical optimal update.

**Set pseudo target $\hat{\boldsymbol{x}}^*$ by GP update:** For the GP update case, we set the pseudo target $\hat{\boldsymbol{x}}^*$ as the one gradient step update using the gradient of the posterior mean prediction of the GP surrogate model as follows:

$$\hat{\boldsymbol{x}}^* = \boldsymbol{x}_K - \nabla \hat{f}(\boldsymbol{x}_K; \boldsymbol{X}^n) \tag{2}$$

where $\boldsymbol{x}_K$ denotes the generated data in Algorithm 2 and $\hat{f}(\boldsymbol{x}_K; \boldsymbol{X}^n)$ denotes the posterior prediction of the GP surrogate model (Seeger, 2004) evaluated at $\boldsymbol{x}_K$, which has closed-form as below:

$$\hat{f}(\boldsymbol{x}_K; \boldsymbol{X}^n) = \boldsymbol{k}(\boldsymbol{x}_K, \boldsymbol{X}^n)^\top \left(\mathcal{K}(\boldsymbol{X}^n, \boldsymbol{X}^n) + \lambda \boldsymbol{I}\right)^{-1} \boldsymbol{y} \tag{3}$$

where $\boldsymbol{X}^n = [\boldsymbol{x}^1, \cdots, \boldsymbol{x}^n]$ and $\boldsymbol{y} = [y^1, \cdots, y^n]$ denotes the collected data and its corresponding score value in the historical set $\mathcal{D}$ in Algorithm 2, respectively. When the historical set $\mathcal{D}$ is empty, we simply set the gradient $\nabla \hat{f}(\boldsymbol{x}_K; \varnothing) = 0$.

In this work, we employ shift-invariant kernels that can be rewritten in the form as $k(\boldsymbol{z}_1, \boldsymbol{z}_2) = g(\|\boldsymbol{z}_1 - \boldsymbol{z}_2\|_2)$, e.g., Gaussian kernel and Matérn kernel. We show that the gradient of the GP surrogate when employing these shift-invariant kernels lies on the low-dimensional data manifold. We present this property in Proposition 1. Detailed proof of Proposition 1 can be found in Appendix E. This property enables us to generate meaningful data on the data manifold instead of degenerated data, which is important for high-dimensional diffusion-model target generation problems, e.g., high-resolution ($1024 \times 1024$) target image generation tasks.

**Proposition 1.** *Given prior data $\boldsymbol{X}^n = [\boldsymbol{x}^1, \cdots, \boldsymbol{x}^n]$ and its corresponding score $\boldsymbol{y} = [y^1, \cdots, y^n]$, let $\hat{f}(\boldsymbol{x}; \boldsymbol{X}^n)$ denotes the posterior mean of the GP model. For shift-invariant kernels $k(\boldsymbol{z}_1, \boldsymbol{z}_2) = g(\|\boldsymbol{z}_1 - \boldsymbol{z}_2\|_2)$, $\nabla \hat{f}(\boldsymbol{x}; \boldsymbol{X}^n)$ lies on a subspace spanned by $[\boldsymbol{x}, \boldsymbol{x}^1, \cdots, \boldsymbol{x}^n]$, where $\boldsymbol{x}^i$ denotes the $i^{th}$ data sample in $\boldsymbol{X}^n$.*

**Remark:** When setting the pseudo target $\hat{\boldsymbol{x}}^*$ as Eq.(2), we know the pseudo target $\hat{\boldsymbol{x}}^*$ lies on the low-dimensional subspace spanned by data according to Proposition 1. Usually, the gradient descent update in Eq.(2) will lead to a new pseudo target $\hat{\boldsymbol{x}}^*$ with a lower score. In addition, based on the empirical observation from Algorithm 1, we know that even the pseudo target $\hat{\boldsymbol{x}}^*$ is noisy, Algorithm 1 can still generate similar data that is meaningful and lies on the data manifold. These two properties enable Algorithm 2 to generate data with lower and lower scores along the data manifold.

**Set pseudo target $\hat{x}^*$ by historical optimal update:** For the historical optimal update case, we set pseudo target $\hat{x}^*$ as the empirical optimal point from the historical dataset $\mathcal{D}$ in Algorithm 2 as:

$$\hat{x}^* = \arg \min_{x \in \mathcal{D}} f(x). \tag{4}$$

Set pseudo target $\hat{x}^*$ as Eq.(4) is an empirical approximation of the optimal target $x^* = \arg \min_{x \in \mathcal{X}} f(x)$. In Algorithm 2, at each iteration w.r.t. the batch query $i$, we employ the generated data that achieves the current best score as the pseudo target $\hat{x}^*$ to guide the generation process. Intuitively, given this pseudo target $\hat{x}^*$, Algorithm 2 will generate a batch of data that is similar to $\hat{x}^*$ on the low-dimensional data manifold, which is promising to further improve the score. By iteratively repeating this procedure, the generated data progressively achieves lower objective scores.

## 4 EXPERIMENTS

In this section, we evaluate our algorithm in two domains, images and molecules, and compare it against four baseline methods: DDPO (Black et al., 2023), DPOK (Fan et al., 2024), D3PO (Yang et al., 2024), and DNO (Tang et al., 2024). Moreover, we further evaluate our algorithm in compressibility, incompressibility, and aesthetic quality tasks in Appendix A.

### 4.1 IMAGE BLACK-BOX TARGET GENERATION TASK

**Problem: Prompt Alignment**. We consider the image-prompt alignment problem. While the current state-of-the-art image generative models excel at generating highly realistic images, they sometimes struggle to faithfully generate images that are accurately aligned with the input prompts, especially those complex prompts involving rare object combinations, object counting, or specific object positioning.

**Pre-trained Model: SDXL-Lightning**. We use the SDXL (Podell et al., 2023) diffusion model as the backbone text-to-image model. It can generate a $1024 \times 1024$ high-resolution realistic image. In our experiment, we use the distilled version, SDXL-Lightning (Lin et al., 2024), for its high sampling efficiency, which can generate images with comparable quality with just $K = 8$ steps. We use the official implementation [2]

**Objective Function: Gemini 1.5**. We leverage Gemini 1.5 (Reid et al., 2024), an advanced multimodal LLM service, as our black-box objective function to evaluate the alignment between input prompts and generated images. To avoid confusion, the term *query* refers to the input to Gemini, while *prompt* refers to the text used for image generation.

The query to Gemini 1.5 consists of the generated images with the question like: "Does the prompt `$prompt` accurately describe the image? Rate from 1 to 5". We state the complete query in Appendix G.

Because it is a closed-source paid service, we limit the number of batch queries in our experiments, referred to as the *batch query budget*. We use the Gemini 1.5 Flash model (code: `gemini-1.5-flash-001`) for its cost-efficiency, and set the temperature to 0 for experiment consistency and reproducibility. For conciseness, we call it simply *Gemini* in the following section.

**Experiment Procedure**. We identify 12 prompts that the pre-trained model SDXL-Lighning struggles to generate, and refer to these as the 12 tasks in our experiment, where the goal is to generate images that are accurately aligned with the input prompts. We compare our Fast Direct algorithm against each baseline method, and each experiment is constrained with a 50 batch query budget.

We perform inference-time guidance using Fast Direct (Algorithm 2) to maximize the Gemini rating. We use the GP model in Eq. 2 for the pseudo-target, and set the kernel as a Gaussian kernel, and we follow (Hvarfner et al., 2024) to set the lengthscale as $\lambda = \sqrt{d}$, where $d = 4 \times 128 \times 128$ is the latent dimensionality of SDXL. We use $N = 50$ iterations to utilize the 50 batch query budget, and set the batch size as $B = 32$ and step size as $\alpha = 80$. We use the `EularDescreteScheduler` (Karras et al., 2022) sampler as suggested by the SDXL-Lightning implementation (Lin et al., 2024), and the `DDIMScheduler` (Song et al., 2020) (DDIM) sampler for a fair comparison with the baselines.

---

[2] https://huggingface.co/ByteDance/SDXL-Lightning

For all baseline methods, we perform 50 iterations, each iteration utilizes one batch query for updating. We set the batch size to 32, and leave the rest of the hyperparameters as default. We state more details for baselines in the Appendix D. Note that for DNO, in which each experiment trial can only produce one image, we run 16 independent experiment trials and report the average results[3].

**Experiment Results**. We present the results for the "deer-elephant" [4] task in this section and defer the results for the other 11 tasks to Appendix J due to space constraints. We compare the images generated by our method with those generated by the baseline methods at each 10 batch query intervals in Fig. 2. We can observe that our algorithm successfully satisfies the visual objectives for both samplers within the 50 batch query budget, while the baseline methods fall short within the same budget. Similar phenomena are observed for the other 11 tasks in Appendix J. We grant baseline methods with extra batch query budget, and we observed that DDPO and DPOK almost achieve the visual objectives, whereas D3PO and DNO remain unsuccessful in this particular task.

We collect the 32 randomly generated images using our algorithm (by utilizing a 50 batch query budget) and present them in Fig. 3. We can observe that most images well satisfy the visual objective. Similar phenomena are observed on the other 11 tasks as shown in Appendix J.

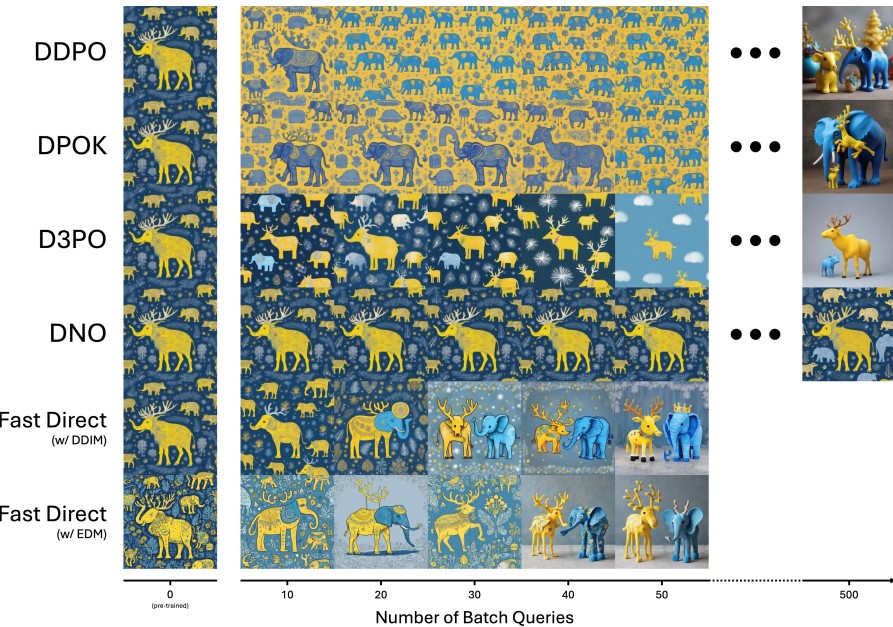

Figure 2: The generated images over each number of batch queries on the prompt "deer-eleplant" [4], extra batch query budget (until 500) is given to the baseline methods for demonstration.

For the quantitative results, we report the average objective values over each batch query across the 12 tasks in Fig. 4. For DNO, which only produces one image each experiment, we average the results from 16 independent experiments. We observe that the Fast Direct achieves nearly a full score within the 50 batch query budget in all tasks, which is significantly higher than the baseline methods. Moreover, we observe that Fast Direct achieve a similar score for both samplers, suggesting it is invariant to the sampler. We further perform an ablation study in Appendix B, and show that Fast Direct is insensitive to the hyper-parameters: step size $\alpha$ and batch size $B$.

We further employ tasks 1 to 3 and assign an extra batch query budget of 200 to the baseline methods. We report the *accumulated objective values*, which represent the best objective values achieved up to each number of batch queries, in Appendix Fig. 11. We observe that our algorithm, utilizing only 50 batch query budget, consistently outperforms all baseline methods even when they are assigned with expanded batch query budget of 200.

To quantify this advantage, in Table 1, we identify the minimum number of batch query budget (denoted as $N^*$) our algorithm requires to surpass each baseline method with their 200 batch query

---

[3]Note that DNO spent $16 \times 50 = 800$ batch queries for each task.

[4]The complete prompt is: `A yellow reindeer and a blue elephant.`

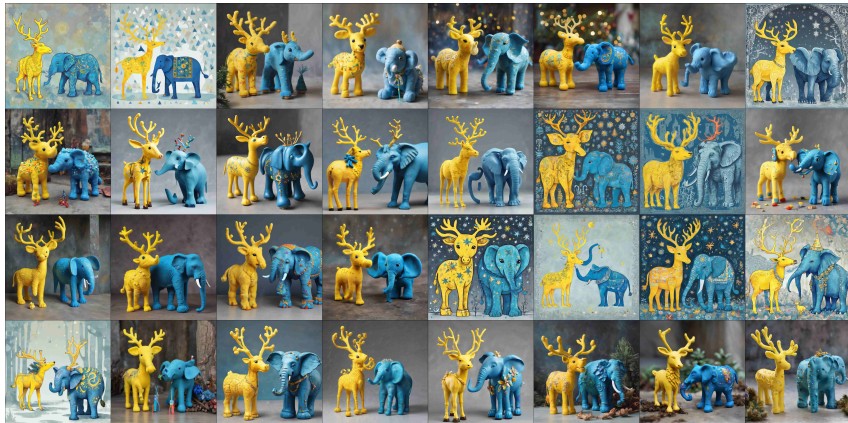

Figure 3: The 32 randomly generated images for the prompt "deer-eleplant" [4] guided by Fast Direct (w/ EDM) by utilizing 50 batch query budget.

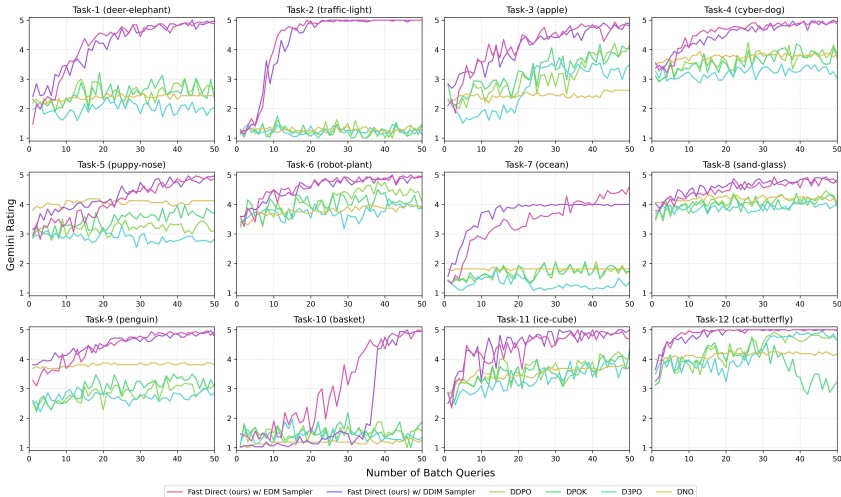

Figure 4: The average Gemini rating (from 1 to 5, higher is better) of the generated images over each number of batch queries on the 12 different tasks (the prompts abbreviation shown in bracket, see the complete prompts in Appendix 3).

budget. We further calculate ours *batch-query-efficiency gain* compared to baselines as $\frac{200}{N^*}$ and report (in the bracket) in Table 1. We can observe that our algorithm is at least 667% and up to 1000% more batch-query-efficient than the baseline methods.

## 4.2 MOLECULE BLACK-BOX TARGET GENERATION TASK

**Problem: Drug Discovery**. We consider the drug discovery problem. One of the key problems is to find drug molecules that have a strong binding affinity with the target protein receptor. The binding affinity quantifies the strength of the interaction between two biomolecules, so a stronger binding affinity indicates a higher drug efficacy.

A traditional approach to this problem requires human-designed molecules, which are highly time-consuming. Recent advancements in AI pose a great potential to speed up this process by generating molecules using advanced generative models such as diffusion models. Although the current pre-trained molecule models, such as TargetDiff, are capable of generating relatively realistic molecules, they cannot generate molecules with the user's desired high binding affinity.

**Pre-trained Model: TargetDiff**. We use TargetDiff (Guan et al., 2023) as the backbone molecules generative model. It is pre-trained on the CrossDocked2020 dataset (Francoeur et al., 2020), and can generate realistic 3D molecular structures conditioned on the given protein receptor. For sam-

Table 1: The minimum number of batch query budget (denoted as $N^*$) for Fast Direct (ours) to outperform each baseline (when each baseline is granted with 200 batch query budget) on images and molecules tasks. The values in brackets show the batch-query-efficiency gain ($\frac{200}{N^*}$) of Fast Direct (ours) over the baselines.

| | Minimum number of batch queries ($N^*$) for **Fast Direct** to outperform each baselines: | | |
| | **DDPO** | **DPOK** | **D3PO** |
|---|---|---|---|
| **Image Tasks** | | | |
| Task 1 | 15 (13.34$\times$) | 15 (13.34$\times$) | 10 (20.00$\times$) |
| Task 2 | 8 (25.00$\times$) | 8 (25.00$\times$) | 4 (50.00$\times$) |
| Task 3 | 37 (5.41$\times$) | 46 (4.34$\times$) | 16 (12.50$\times$) |
| Average | 20 (10.00$\times$) | 23 (8.70 $\times$) | 30 (6.67$\times$) |
| **Molecules Tasks** | | | |
| Task 1 | 6 (33.33$\times$) | 5 (40.00$\times$) | 25 (8.00$\times$) |
| Task 2 | 9 (22.22$\times$) | 4 (50.00$\times$) | 11 (18.18$\times$) |
| Average | 7.5 (26.67$\times$) | 4.5 (44.44$\times$) | 18 (11.11$\times$) |

pling efficiency, we use the `DDIMScheduler` sampler with $K = 200$ sampling steps as we find that it can generate comparable results. During the generation process, we fix the atom type according to the dataset reference and only allow the atom position to be varied. We use the official implementation [5] in our experiment.

**Objective Function: Molecules Binding Affinity.** Measurement of the binding affinity requires a real-world experiment, which is highly expensive (David et al., 2020). In our experiment, we leverage the Vina score (kcal/mol) calculated by the AutoDock Vina simulation software (Eberhardt et al., 2021) to estimate the binding affinity. A lower (more negative) Vina score indicates a stronger estimated binding affinity. Therefore, our objective is to minimize the Vina score.

**Experiment Procedure.** We conduct experiments on 6 tasks, where the goal of each task is to optimize the Vina score on the protein receptors of ID from 1 to 6, respectively. Similar to image tasks, each algorithm is constrained to 50 batch query budget.

We perform inference-time guidance using our Algorithm 2 to minimize the Vina score. We use the historical optimal in Eq. 4 for the pseudo-target. We use $N = 50$ iterations to utilize the 50 batch query budget, and set the batch size as $B = 32$, and the step size as $\alpha = 10^{-2}$. We then conduct experiments on the baseline methods for 50 fine-tuning iterations. For DDPO and DPOK, the batch size is set as the same 32; while D3PO requires binary rewards, so we double the batch size as 64 [6]. We keep the hyperparameters as default for all baseline methods.

**Experiment Results**. We report the average Vina score (recall, lower is better) for each batch of queries across 6 tasks in Fig. 10. We observe that our algorithm consistently achieves much lower Vina score compared to the baseline methods across all tasks. Similar to the images task, we grant extra batch query budget of 200 to the baselines for tasks 1 and 2, and report the *accumulated objective values* in Appendix Fig 12, and list the *batch-query-efficiency gain* in Table 1. We observe that our algorithm is at least 1111% and up to 4444% more batch-query-efficient compared to the baseline methods.

## 5 CONCLUSION

In this work, we proposed **Fast Direct** for diffusion model target generation, which effectively addresses the challenges of limited batch query budgets and black-box objective functions, demonstrating its potential for various applications, including image generation and drug discovery. Fast Direct is highly practical, as it is easy to implement, supports any type of SDE solver, and has only one hyper-parameter to tune (step size $\alpha$). Our algorithm is based on the surprising empirical observation that the *universal update direction* (i.e., $\boldsymbol{x}^* - \boldsymbol{x}_K$ in Algorithm 1) can efficiently guide the diffusion trajectory toward the target, even when the target $\boldsymbol{x}^*$ is extremely noisy. This phenomenon suggests an intriguing robustness in the diffusion inference process. Future research could investigate its underlying theoretical principles.

---

[5] https://github.com/guanjq/targetdiff

[6] However, unlike image tasks where Gemini can evaluate rewards for two images in a single query, the Vina simulation lacks this capability, requiring 64 individual queries for a batch of size 64.

ACKNOWLEDGMENTS

This research is supported by the National Research Foundation, Singapore and Infocomm Media Development Authority under its Trust Tech Funding Initiative, the Center for Frontier Artificial Intelligence Research Center, Institute of High Performance Computing, A*Star, and the College of Computing and Data Science at Nanyang Technological University. Any opinions, findings and conclusions or recommendations expressed in this material are those of the author(s) and do not reflect the views of National Research Foundation, Singapore and Infocomm Media Development Authority. Yueming Lyu is supported by Career Development Fund (CDF) of the Agency for Science, Technology and Research (A*STAR) (No: C243512014 and No: C233312007).

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

# A MORE EXPERIMENTS: COMPRESSIBILITY, INCOMPRESSIBILITY, AND AESTHETIC QUALITY

**Problem.** We follow DDPO (Black et al., 2023) to evaluate our algorithm on the three black-box optimization tasks for images: compressibility, incompressibility, and aesthetic quality.

**Objective Function.** For compressibility, we aimed to minimize the compressed JPEG size (MB) of the generated images. For incompressibility, it's simply the inverse. For aesthetic quality, the aesthetic score is evaluated by the pre-trained LAION aesthetics predictor (**?**), which is trained on human ratings of the images' aesthetic quality, and we aimed to maximize the aesthetic score.

**Experiment Procedure.** For each task, we uniformly sample from the 45 common animals (as proposed by DDPO (Black et al., 2023)) as input prompts and optimize the objective using Fast Direct in Algorithm 2. The prompt for each instance within a batch is sampled randomly and independently. We set $N = 50$ batch query budget for compressibility and incompressibility, and $N = 100$ for aesthetic quality. We use the same pre-trained model, hyper-parameters, and GP settings as in Section 4.1. The 45 common animal prompts used in the experiment are as follows:

| cat | dog | horse | monkey | rabbit | zebra | spider | bird | sheep |
|------|--------|---------|-----------|--------|---------|---------|---------|----------|
| deer | cow | goat | lion | tiger | bear | raccoon | fox | wolf |
| lizard | beetle | ant | butterfly | fish | shark | whale | dolphin | squirrel |
| mouse | rat | snake | turtle | frog | chicken | duck | goose | bee |
| pig | turkey | fly | llama | camel | bat | gorilla | hedgehog | kangaroo |

For DNO, each experimental trial generates only a single image, and the objective score is highly dependent on the specific prompt. To ensure a more accurate evaluation, we run 45 independent experiment trials using each prompt and report the average results. Consequently, DNO requires $45\times$ more batch queries per task compared to other methods.

**Experiment Result.** We present the objective scores for each task in the left column of Fig. 5 and provide the generated images for the three tasks in the supplementary materials. For compressibility and incompressibility, we observe that Fast Direct achieves significantly better scores than the baselines. For aesthetic quality, Fast Direct with the EDM sampler achieves significantly better scores than with DDIM, likely due to EDM being a more advanced sampler, capable of generating higher-quality images. DNO achieves comparable scores; however, note that each experiment optimizes and generates only a single image.

**Generalization Ability.** We follow DDPO to evaluate generalization ability. For Fast Direct, we freeze the learned GP model to generate 16 unseen images using distinct unseen animal prompts. Specifically, in this phase, Lines 17 and 18 are removed from Algorithm 2, and our algorithm does not access the objective function. For DDPO, DPOK, and D3PO, the fine-tuned models are frozen to generate images with the 16 unseen animal prompts. DNO is not applicable to the generalization experiment, as it performs per-image optimization and lacks generalization ability. For the unseen prompts, since DDPO didn't publish the unseen prompts, we created our own as follows:

| elephant | eagle | pigeon | hippo | hamster | otter | panda | reindeer |
|----------|---------|----------|-------|---------|--------|--------|----------|
| owl | penguin | flamingo | seal | koala | giraffe | parrot | cheetah |

In Fig.6, we present the images generated using the unseen prompt "hippo" by each algorithm across the three tasks. The remaining 15 prompts for the three tasks are provided in the supplementary material. A similar phenomenon is observed for all 15 other unseen prompts across the three tasks. For quantitative evaluation, we report the objective values for the unseen prompts in the right column of Fig.5. We can observe that Fast Direct with the GP model has generalization ability to unseen prompts.

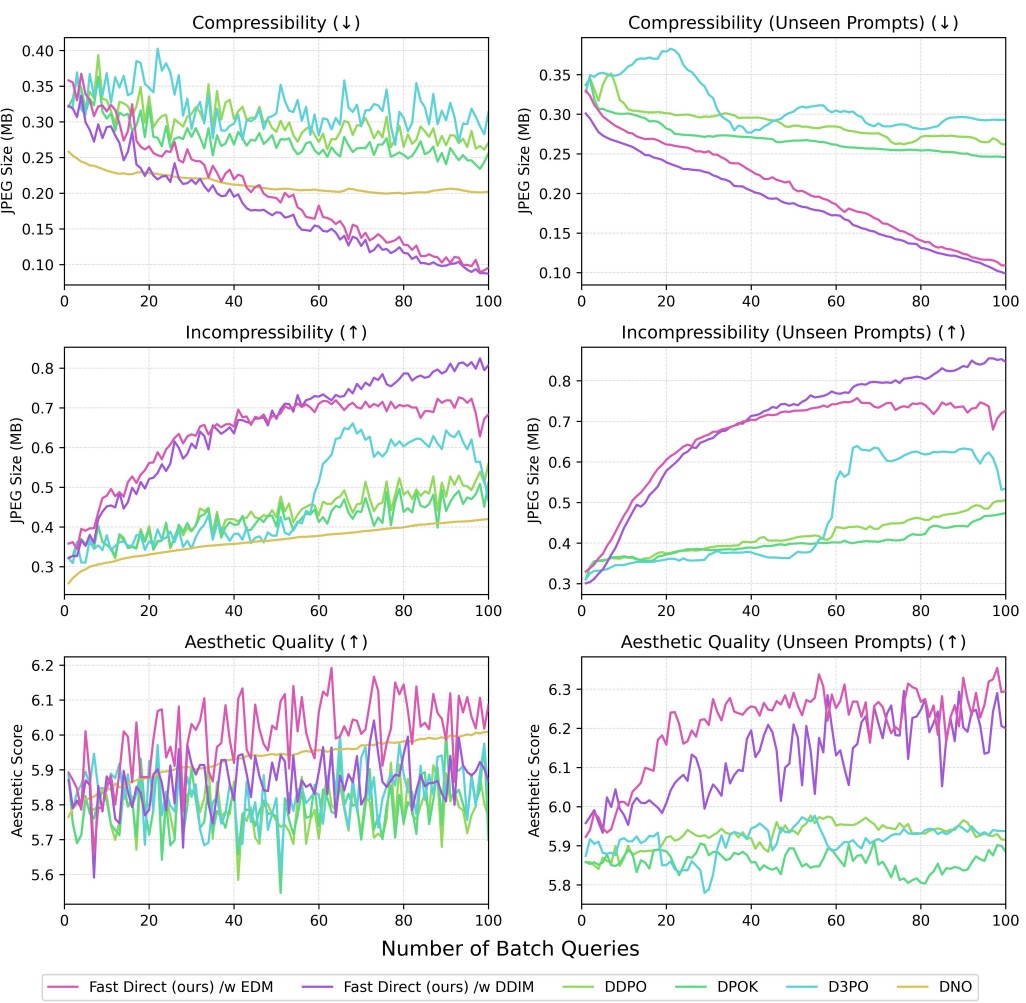

Figure 5: Left column: The average objective score of the generated images over each number of batch queries on the 3 black-box optimization tasks, the images are generated using the 45 common animals that were used in DDPO (Tang et al., 2024) as the input prompts. Right column: The objective score of the images generated by **unseen prompts**, which demonstrates the generalization capability. Note that DNO is not applicable to this task.

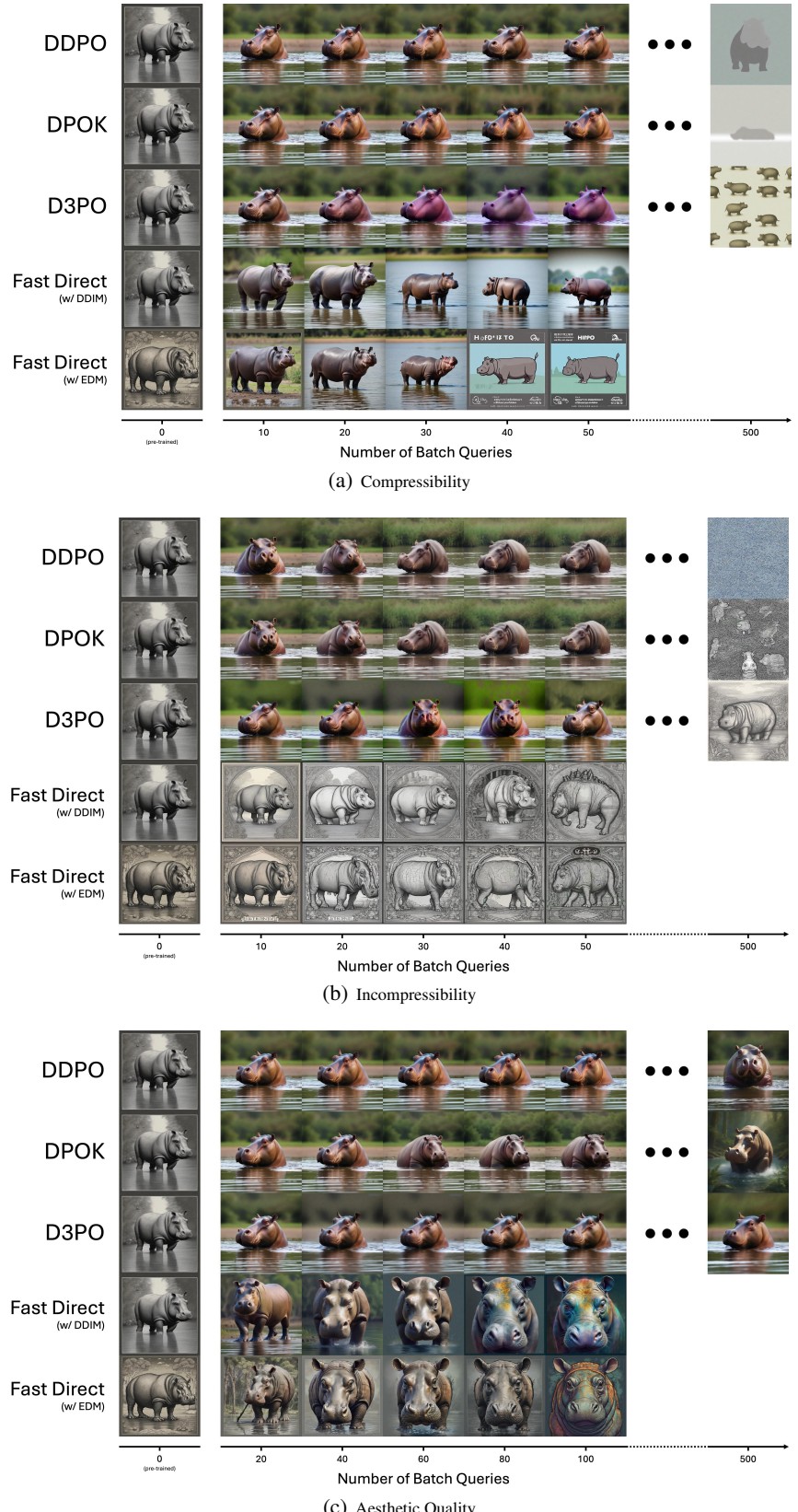

Figure 6: The generated images using the unseen prompt "hippo" in the aesthetic quality task, extra batch query budgets (until 500) are given to the baseline methods for demonstration.

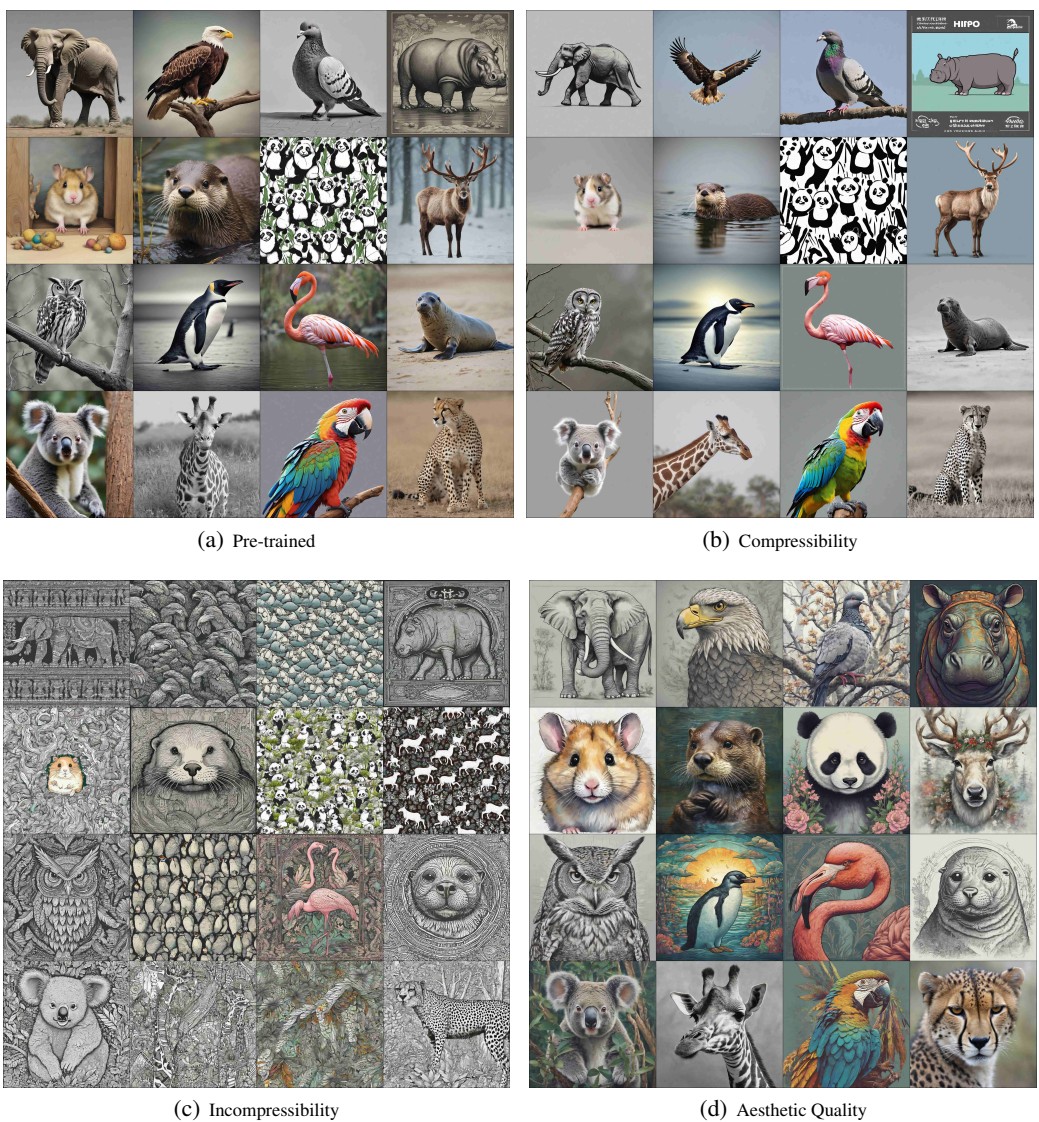

(a) Pre-trained

(b) Compressibility

(c) Incompressibility

(d) Aesthetic Quality

Figure 7: The generated images using the 16 unseen prompts. Top left is the image generated by the pre-trained model, the rest are the Fast Direct generated images in the corresponding tasks.

# B  ABLATION STUDY

We perform an ablation study on the target image generation Task-1. For step size analysis, we fix the batch size as 32, then perform experiment with different step size $\alpha = \{20, 40, 80, 160, 320\}$; for batch size analysis, we fix the step size as $80$, then perform experiment with different batch size $B = \{4, 8, 16, 32, 64\}$. Additionally, we report the run time for different batch sizes. We report the result in Fig. 8.

We can observe that the performance steadily increases for any step size and any batch size, which suggests that our algorithm is not sensitive to the hyperparameter settings. The run time scales linearly with the batch size. In our target image generation experiments in Section 4.1, as the batch size is set as 32, each experiment takes approximately $6.4$ hours to process 32 images in parallel.

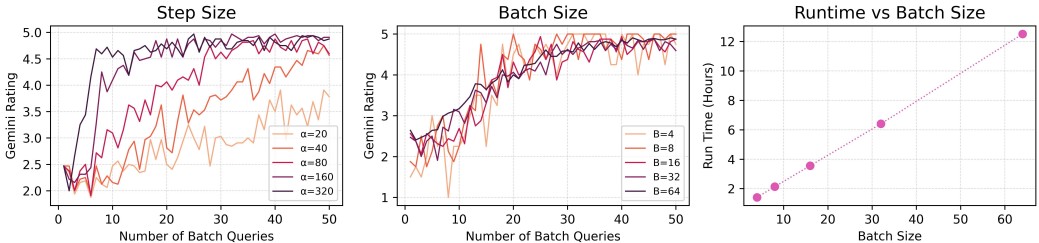

Figure 8: Left: Gemini rating for different step size settings. Middle: Gemini rating for different batch size settings. Right: Run time (hours) for different batch size settings.

# C  ANALYSIS OF UNIVERSAL DIRECTION

In Fig. 9, we demonstrate Algorithm 1 with update direction $\vec{d} = \boldsymbol{x}^* - \boldsymbol{x}_{K'}$ for $K' \in \{K, K/2, K/4, K/8\}$. It shows that the generated image quality decreases as the $K'$ decreases. This is because as $K'$ decreases, the $\boldsymbol{x}_{K'}$ becomes noisier and may move further away from the data manifold.

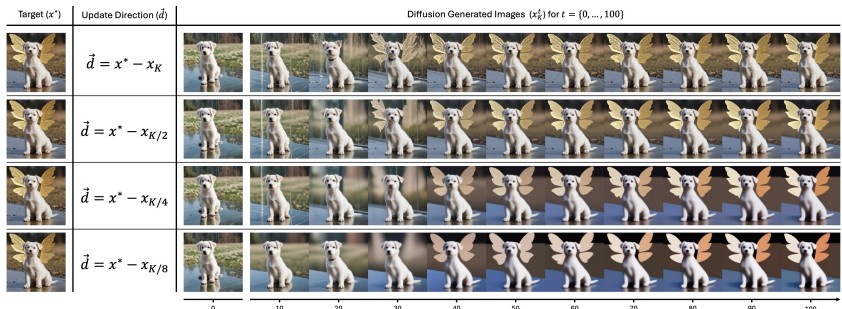

Figure 9: The update direction $\vec{d} = \boldsymbol{x}^* - \boldsymbol{x}_{K'}$ for $K' \in \{K, K/2, K/4, K/8\}$, the generated images quality decrease as the $\boldsymbol{x}_{K'}$ being more more noisy.

# D  BASELINES DETAILS

For DDPO, DPOK, and D3PO, we fine-tune the model to maximize the Gemini rating. We fine-tune the model with $50$ epochs; each epoch utilizes one batch query for model updating. For DDPO and DPOK, we set the batch size to 32; for D3PO, it requires a relative reward, so we doubled the batch size to $64$.[7] As these RL-based methods require closed-form expression of the logarithm probabilities, we follow their official implementation to use the `DDIMScheduler` (Song et al., 2020) sampler.

---

[7]For D3PO, to evaluate $64$ images only requires 32 queries calls, see Appendix G.

For DNO, we optimize the noise sequence w.r.t the Gemini rating. Recall that the Gemini is a black-box, so we use the non-differentiable mode, with the number of samples for gradient approximation set as 32, and optimize for 50 iterations; each iteration utilizes one batch query for updating. However, each experiment of DNO only produces one image, where the score is highly dependent on the initial prior. Thus, for a more accurate evaluation, we run 16 independent experiments to generate 16 images and report the average results. Note that this requires $16 \times 50 = 800$ batch queries for each task, which is 16 times compared with our Fast Direct and other baselines.

## E  PROOF OF PROPOSITION 1

*Proof.* Let $\boldsymbol{\alpha} = \left(\mathcal{K}(\boldsymbol{X}^n, \boldsymbol{X}^n) + \lambda \boldsymbol{I}\right)^{-1} \boldsymbol{y} = [\alpha_1, \cdots, \alpha_n]^\top$, for GP with a shift-invariant kernel that can be rewritten as $k(\boldsymbol{z}_1, \boldsymbol{z}_2) = g(\|\boldsymbol{z}_1 - \boldsymbol{z}_2\|_2)$, the gradient of the GP prediction is

$$\nabla \hat{f}(\boldsymbol{x}; \boldsymbol{X}^n) = \nabla \boldsymbol{k}(\boldsymbol{x}, \boldsymbol{X}^n)^\top \boldsymbol{\alpha} \tag{5}$$

$$= \sum_{i=1}^n \alpha_i \nabla k(\boldsymbol{x}, \boldsymbol{x}^i) \tag{6}$$

$$= \sum_{i=1}^n \frac{\alpha_i}{\|\boldsymbol{x} - \boldsymbol{x}^i\|} g'(\|\boldsymbol{x} - \boldsymbol{x}^i\|_2)(\boldsymbol{x} - \boldsymbol{x}^i) \tag{7}$$

$$= \sum_{i=1}^n c_i(\boldsymbol{x})(\boldsymbol{x} - \boldsymbol{x}^i) \tag{8}$$

where $\boldsymbol{x}^i$ denotes the $i^{th}$ sample in $\boldsymbol{X}^n = [\boldsymbol{x}^1, \cdots, \boldsymbol{x}^n]$, and $c_i(\boldsymbol{x}) = \frac{\alpha_i}{\|\boldsymbol{x} - \boldsymbol{x}^i\|} g'(\|\boldsymbol{x} - \boldsymbol{x}^i\|_2)$, and $g'(\cdot)$ denotes the derivative of $g(\cdot)$.

From Eq.(8), we can see that the gradient $\nabla \hat{f}(\boldsymbol{x}; \boldsymbol{X}^n)$ is a weighted sum of $\boldsymbol{x} - \boldsymbol{x}^i$ for $i \in \{1, \cdots, n\}$. Thus, we know $\nabla \hat{f}(\boldsymbol{x}; \boldsymbol{X}^n)$ lies on a subspace spanned by $[\boldsymbol{x}, \boldsymbol{x}^1, \cdots, \boldsymbol{x}^n]$

$\square$

## F  OVERVIEW OF RELATED WORKS

We present the summary of related works according to their category and supported problem scenarios in Table 2.

Table 2: Overview of existing approaches.

| Algorithm Category | Algorithm | Online | Black-box |
|---|---|---|---|
| Training / Fine-tune | DRaFT (Clark et al., 2023) | ✓ | ✗ |
| | DDOM (Krishnamoorthy et al., 2023), RCGDM (Yuan et al., 2024), DPO (Wallace et al., 2024) | ✗ | ✓ |
| | DDPO (Black et al., 2023), DPOK (Fan et al., 2024), D3PO (Yang et al., 2024) | ✓ | ✓ |
| Inference-time Guidance | LGD (Song et al., 2023), MPGD (He et al., 2023), DNO (2024a) (Karunratanakul et al., 2024), ReNO (Eyring et al., 2024) | ✓ | ✗ |
| | CEP (Lu et al., 2023) | ✗ | ✓ |
| | DNO (Tang et al., 2024), CASBO (Lyu et al., 2024), **Fast Direct** (ours) | ✓ | ✓ |

## G  EXPERIMENT DETAILS

We use the following query question for the Gemini:

```
Does the prompt $prompt accurately describe the image?
Rate from 1 (inaccurate) to 5 (accurate).
Answer in the format:  Score=(score), Reason=(reason).
```

where the `$prompt` is substituted by the input prompt used for image generation. We extract the integer score as the objective value.

The D3PO takes relative reward, so we specially crafted for its selective query question:

```
Given this two images, which image is better aligned with
the prompt $prompt?
Answer in the format:  Choice=(1/2), Reason=(reason).
```

where we extract the choice (1/2) as the selections.

The Gemini 1.5 Flash can faithfully respond with the correct format to ensure our experiment is consistent.

## H   MOLECULES GENERATION EXPERIMENT RESULTS

We report the average Vina score in the molecules generation task in Fig. 10.

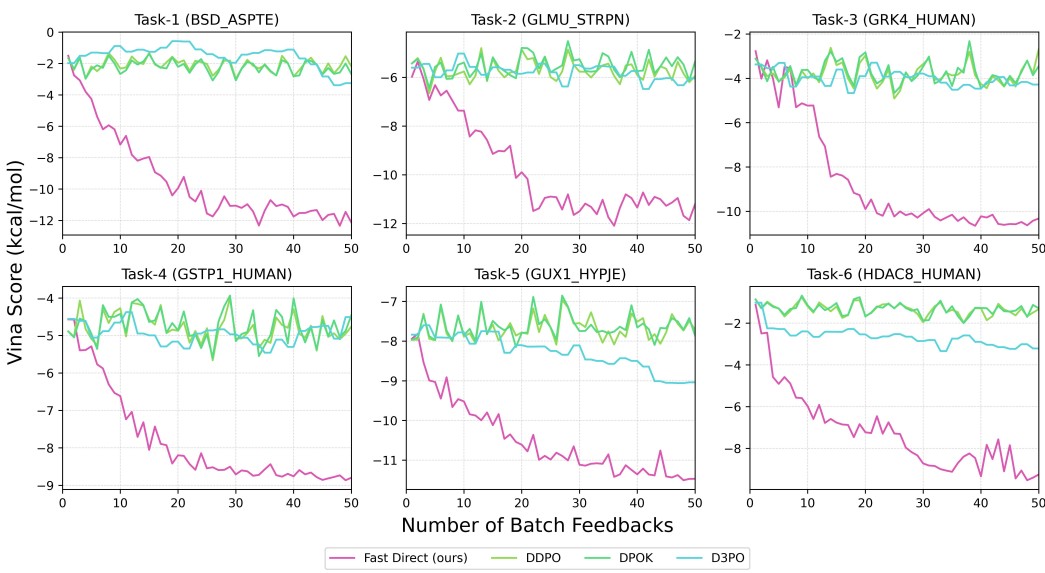

Figure 10: The Vina score (lower is better) of the generated molecules for each number of batch queries on the six protein receptors.

## I   MORE EXPERIMENT RESULTS

We report the accumulated objective values over the number of batch queries for the images task in Fig. 11, and for the molecules task in Fig. 12.

## J   MORE GENERATED IMAGES BY FAST DIRECT FOR IMAGE-PROMPT ALIGNMENT TASK

Table 3 shows the list the complete prompts used for each task in our image generation experiment. From Fig. 13 to Fig. 23, we present the generated images of each algorithm over each number of batch queries for prompts 2 to 12. From Fig. 24 to Fig. 34, we showcase the 32 randomly generated by the pre-trained model, followed by the resultant images guided by the Fast Direct for prompts 2 to 12.

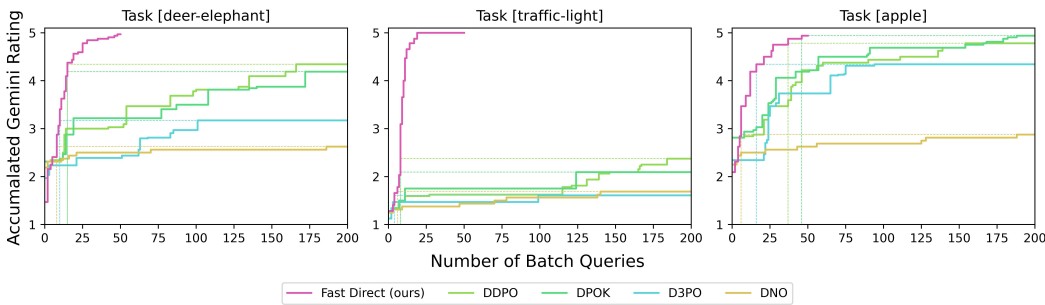

Figure 11: The accumulated Gemini rating (from 1 to 5, higher is better) over the number of batch queries. The accumulated plot displays the maximum Gemini rating achieved for each number of batch queries.

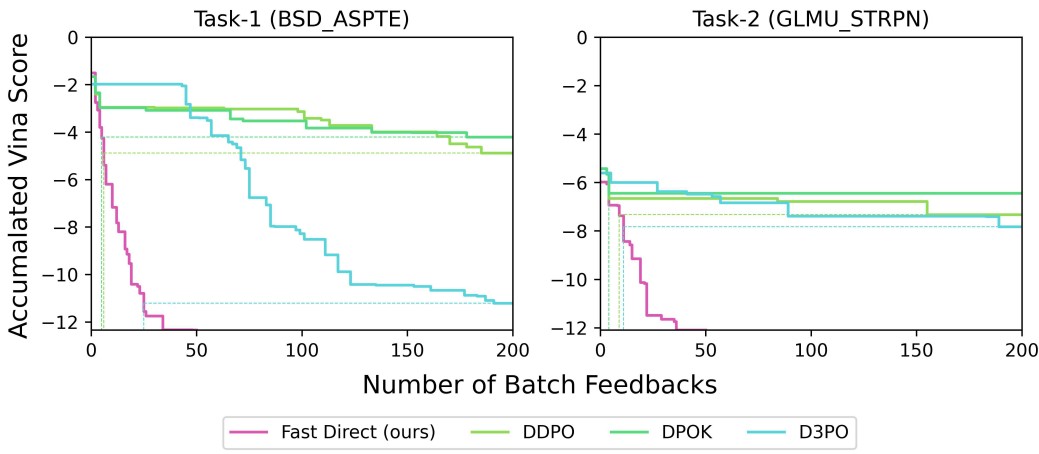

Figure 12: The accumulated Vina score (lower is better) over the number of batch queries. The accumulated plot displays the minimum Vina score achieved up to each number of batch queries.

Table 3: Target Image Generation Tasks

| Task ID | Abbreviated Prompt | Complete Prompt |
|---|---|---|
| 1 | deer-elephant | A yellow reindeer and a blue elephant. |
| 2 | traffic-light | A traffic light with yellow at top, green at middle, and red at bottom. |
| 3 | apple | Seven red apples arranged in a circle. |
| 4 | cyber-dog | A natural fluffy dog talking to a cybertic dog. |
| 5 | puppy-nose | Side view of a puppy lying on floor, one butterfly stopping on its nose. |
| 6 | robot-plant | A cute cybernetic robot plants a tree in the forest. |
| 7 | ocean | A helicopter floating under the ocean. |
| 8 | sand-glass | A transparent glass filled with a mixture of water and sand, with one feather floating inside. |
| 9 | penguin | A photo realistic photo showing a penguin standing on a very small floating ice, with a tree is on fire in the background. |
| 10 | basket | Exactly one orange in a basket of apples. |
| 11 | ice-cube | A glass of water with exactly one ice cube. |
| 12 | cat-butterfly | A cat with butterfly wings. |

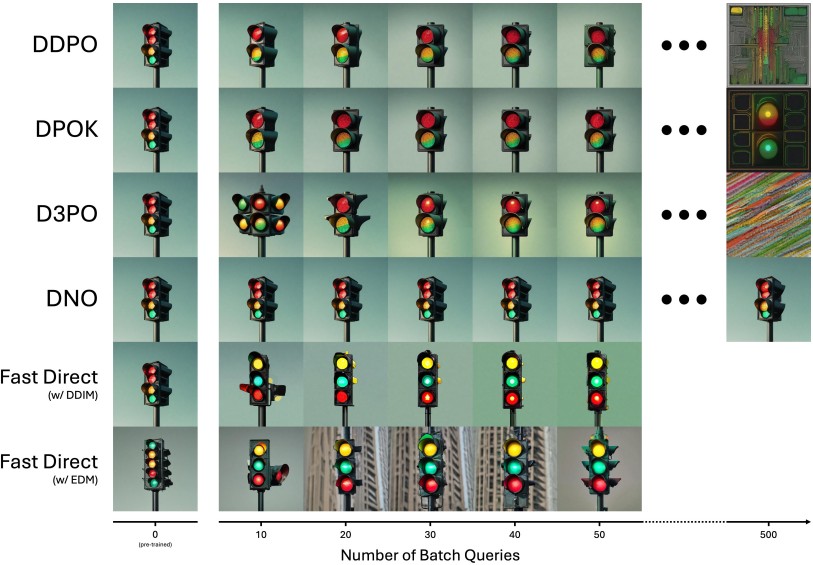

Figure 13: The generated images over each number of batch queries on the prompt "traffic-light", extra batch query budgets (until 500) are given to the baseline methods for demonstration.

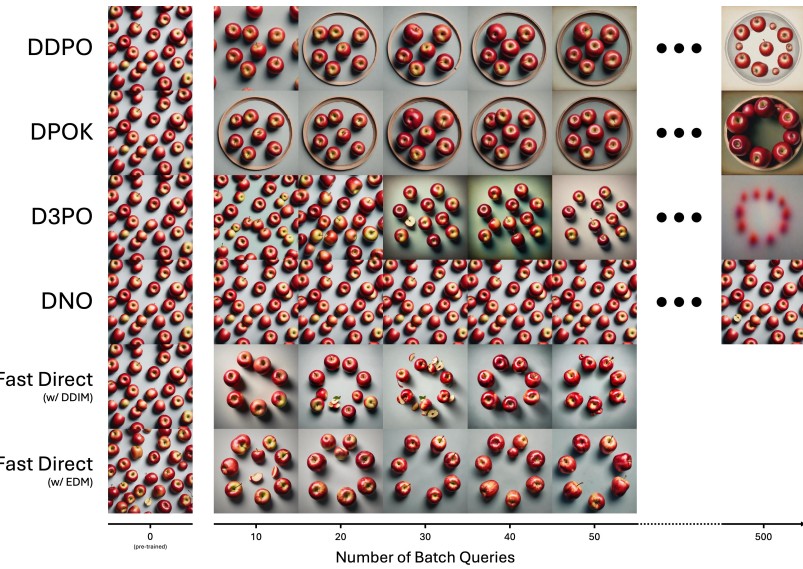

Figure 14: The generated images over each number of batch queries on the prompt "apple", extra batch query budgets (until 500) are given to the baseline methods for demonstration.

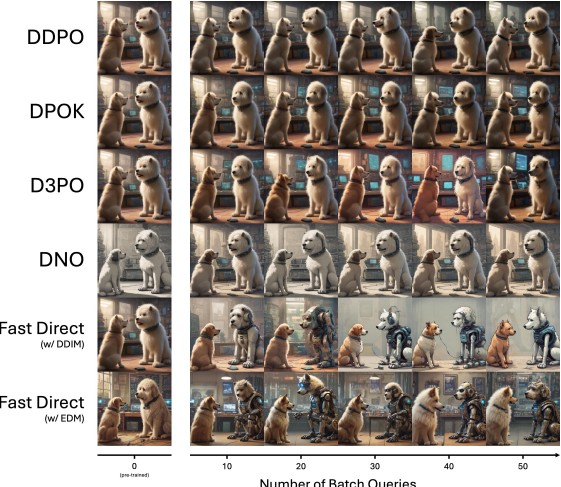

Figure 15: The generated images over each number of batch queries on the prompt "cyber-dog" for each algorithm.

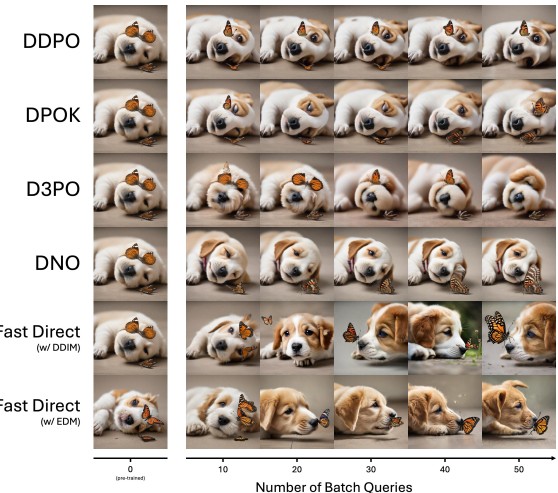

Figure 16: The generated images over each number of batch queries on the prompt "puppy-nose" for each algorithm.

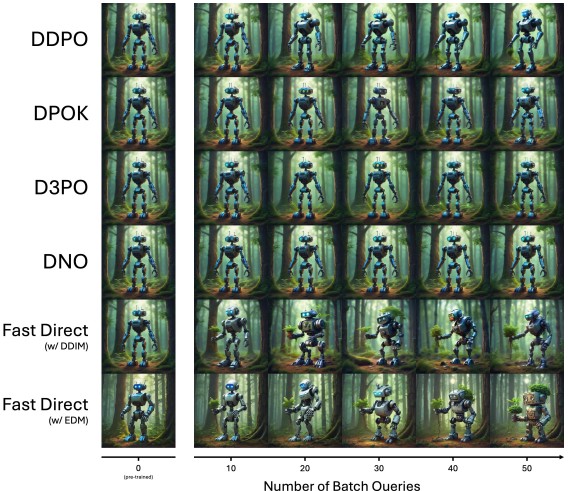

Figure 17: The generated images over each number of batch queries on the prompt "robot-plant" for each algorithm.

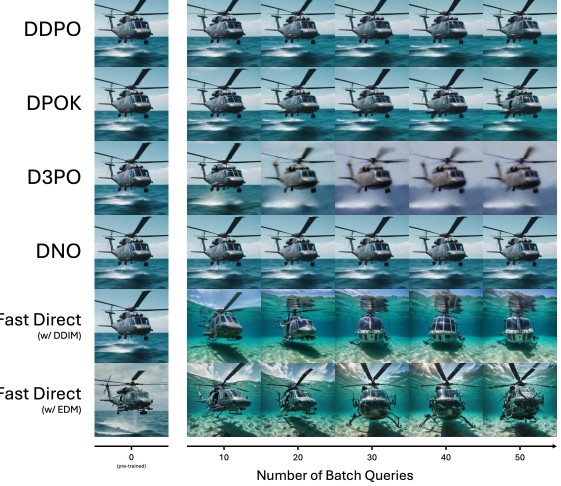

Figure 18: The generated images over each number of batch queries on the prompt "ocean" for each algorithm.

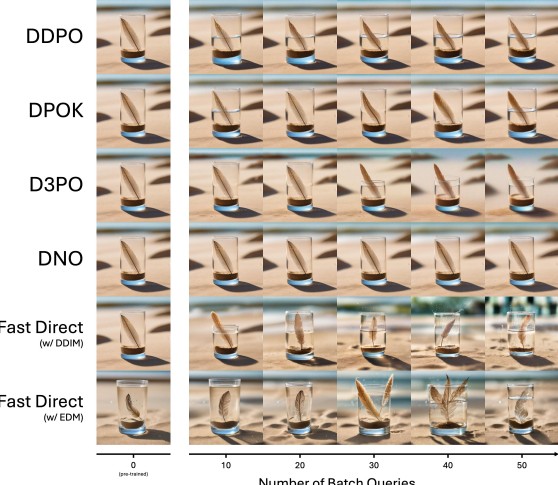

Figure 19: The generated images over each number of batch queries on the prompt "sand-glass" for each algorithm.

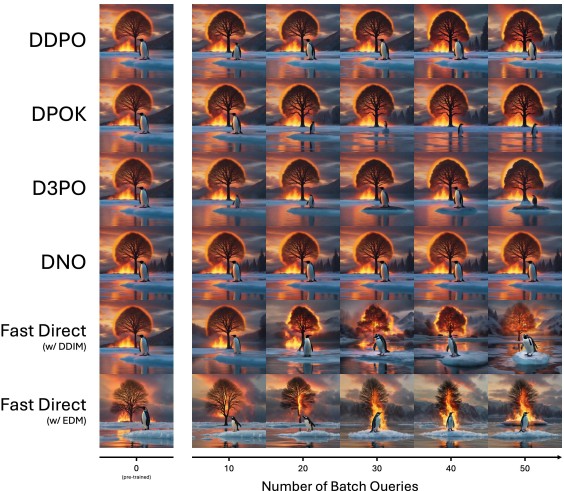

Figure 20: The generated images over each number of batch queries on the prompt "penguin" for each algorithm.

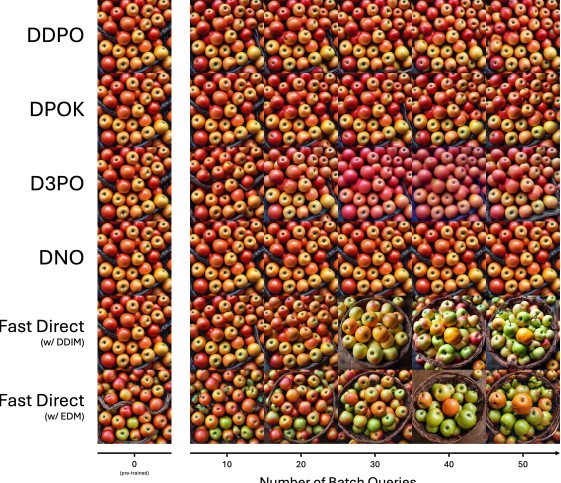

Figure 21: The generated images over each number of batch queries on the prompt "basket" for each algorithm.

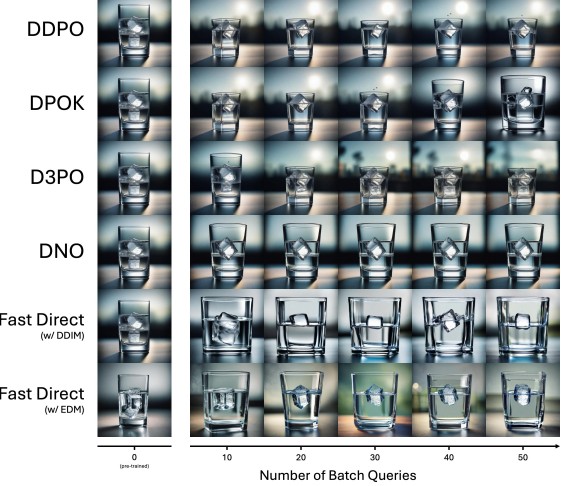

Figure 22: The generated images over each number of batch queries on the prompt "ice-cube" for each algorithm.

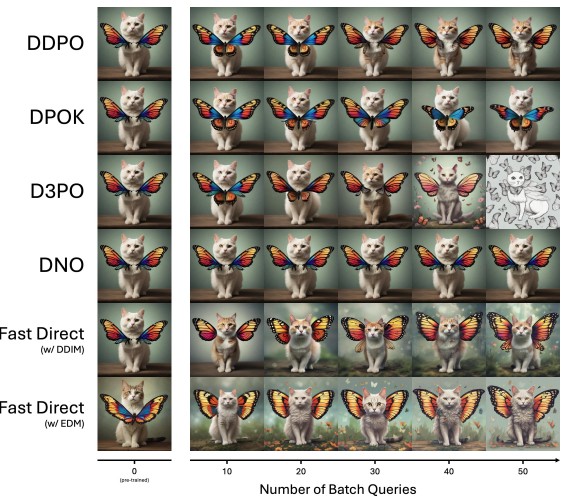

Figure 23: The generated images over each number of batch queries on the prompt "cat-butterfly" for each algorithm.

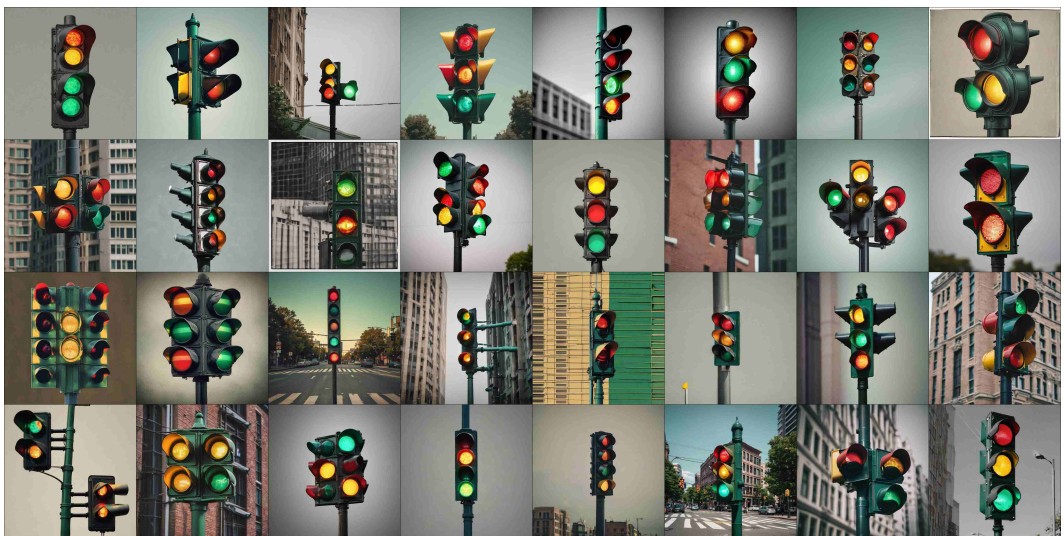

(a) Pre-trained

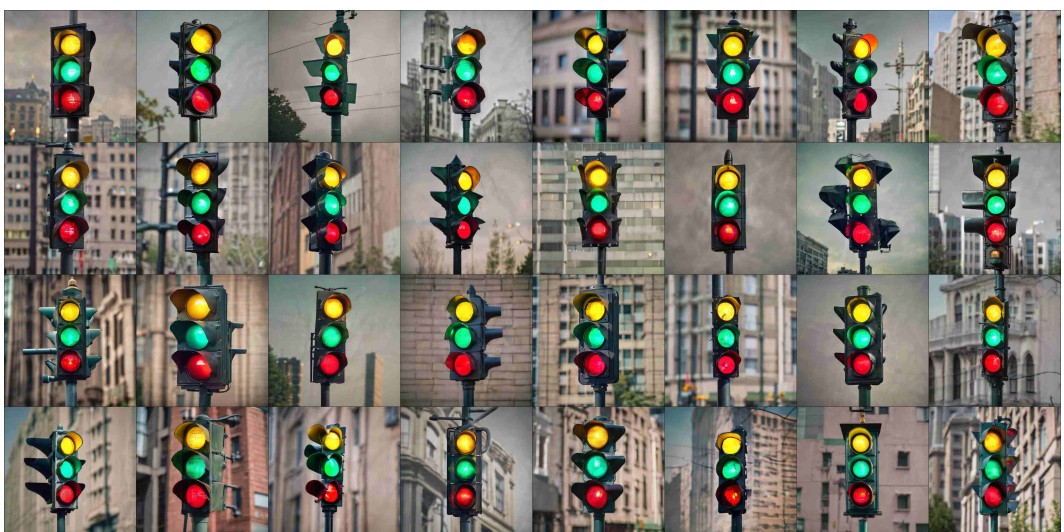

(b) Fast Direct

Figure 24: The 32 randomly generated images for the prompt "traffic-light" guided by Fast Direct by utilizing 50 batch query budget.

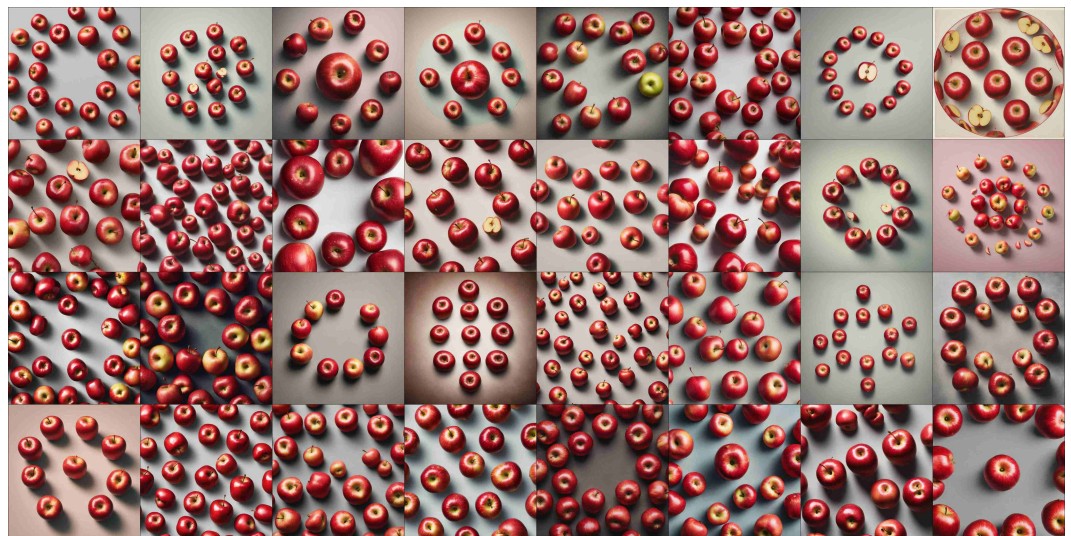

(a) Pre-trained

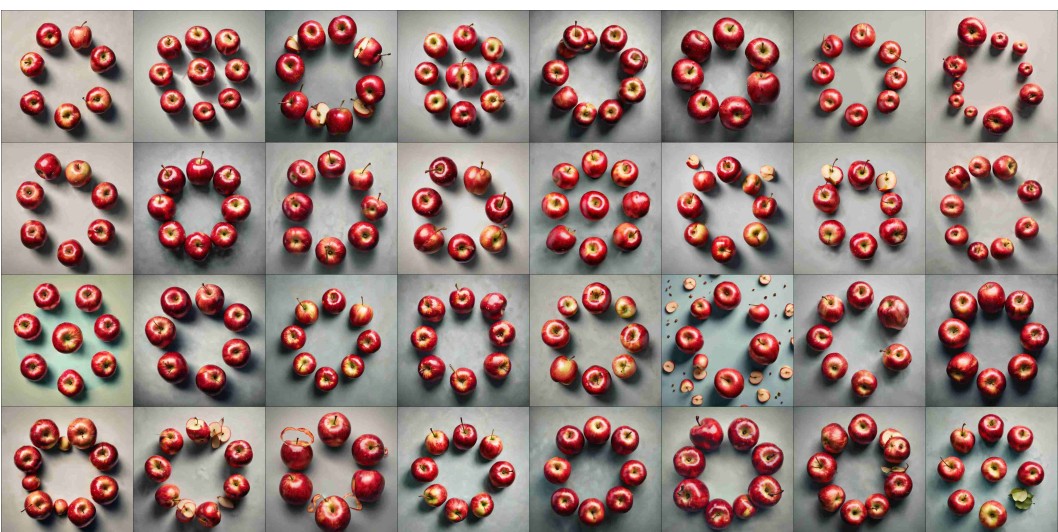

(b) Fast Direct

Figure 25: The 32 randomly generated images for the prompt "apple" guided by Fast Direct by utilizing 50 batch query budget.

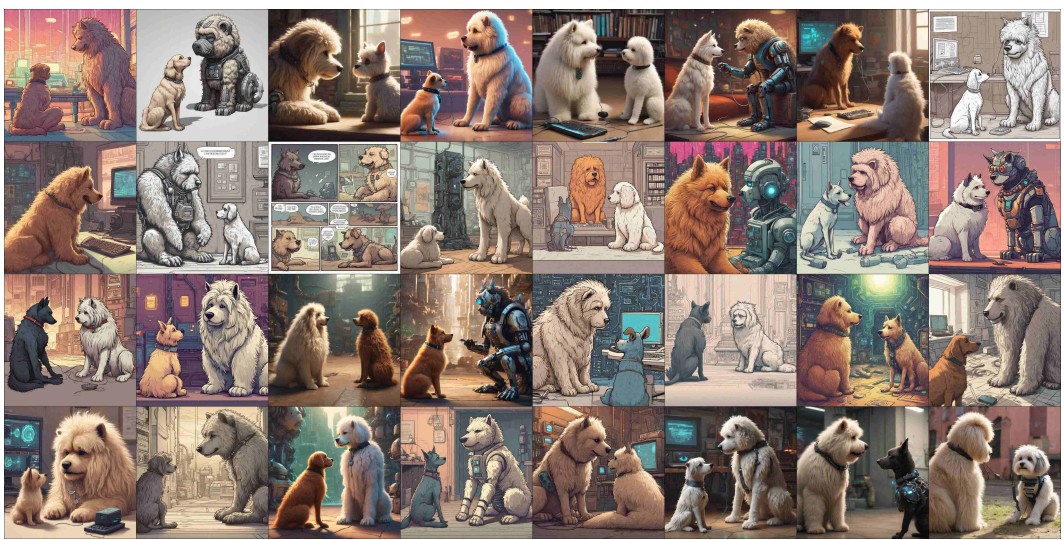

(a) Pre-trained

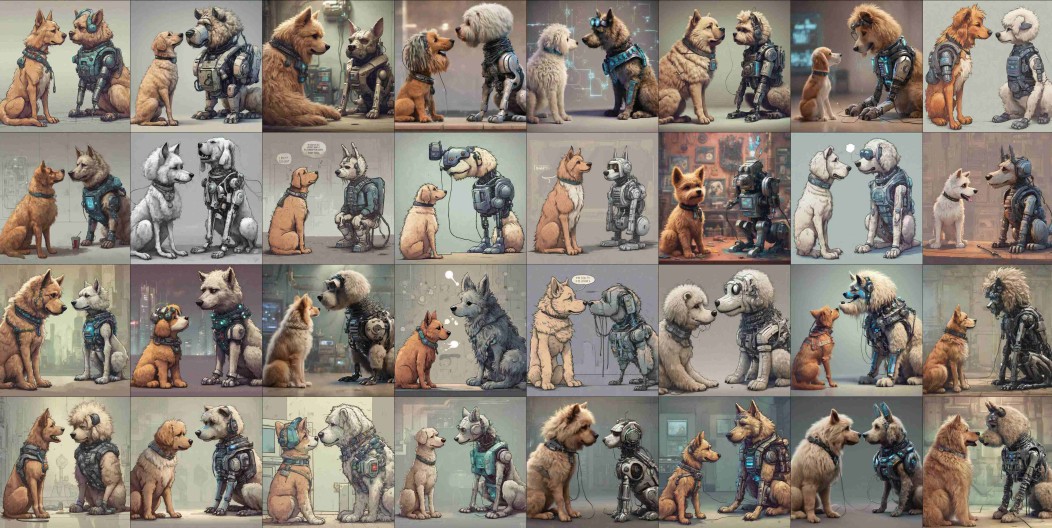

(b) Fast Direct

Figure 26: The 32 randomly generated images for the prompt "cyber-dog" guided by Fast Direct by utilizing 50 batch query budget.

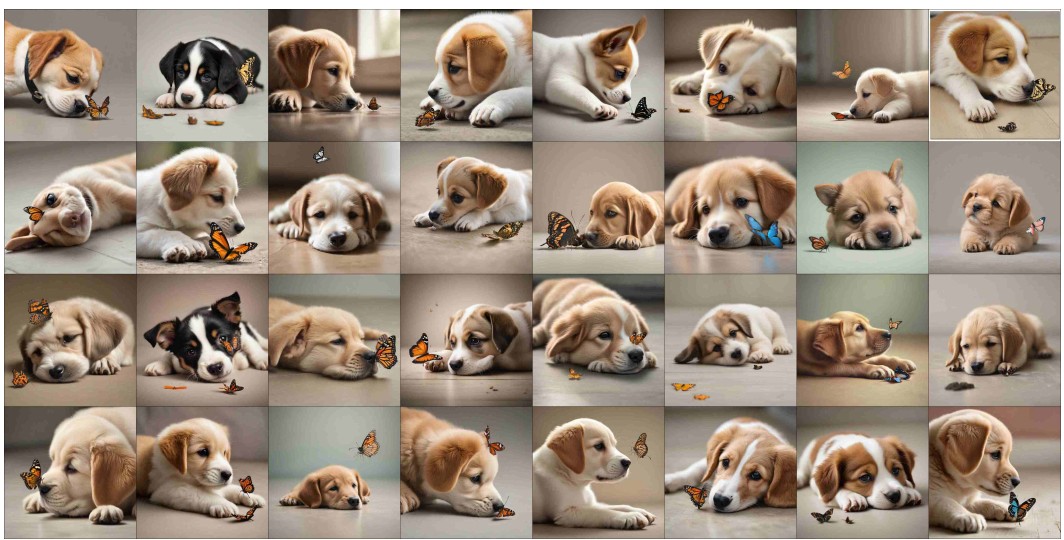

(a) Pre-trained

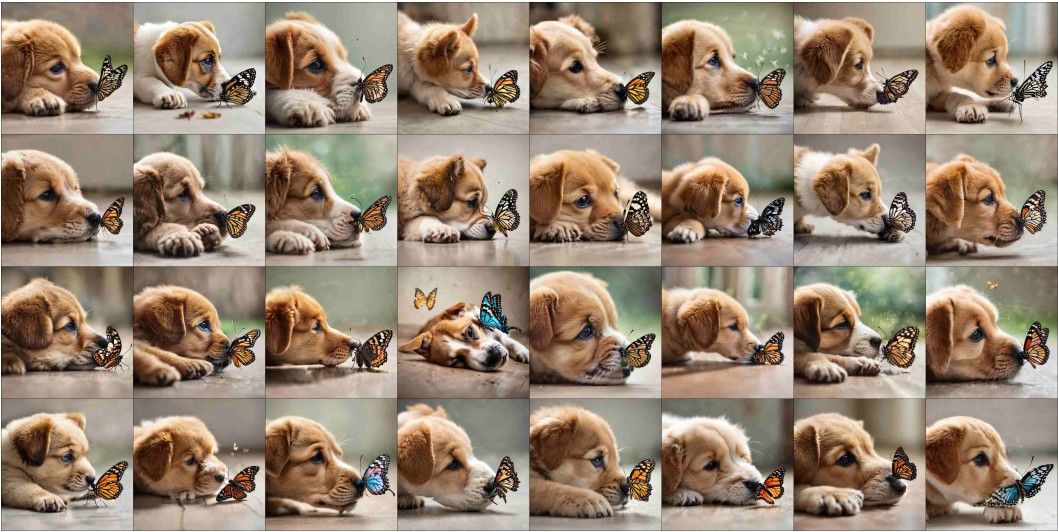

(b) Fast Direct

Figure 27: The 32 randomly generated images for the prompt "puppy-nose" guided by Fast Direct by utilizing 50 batch query budget.

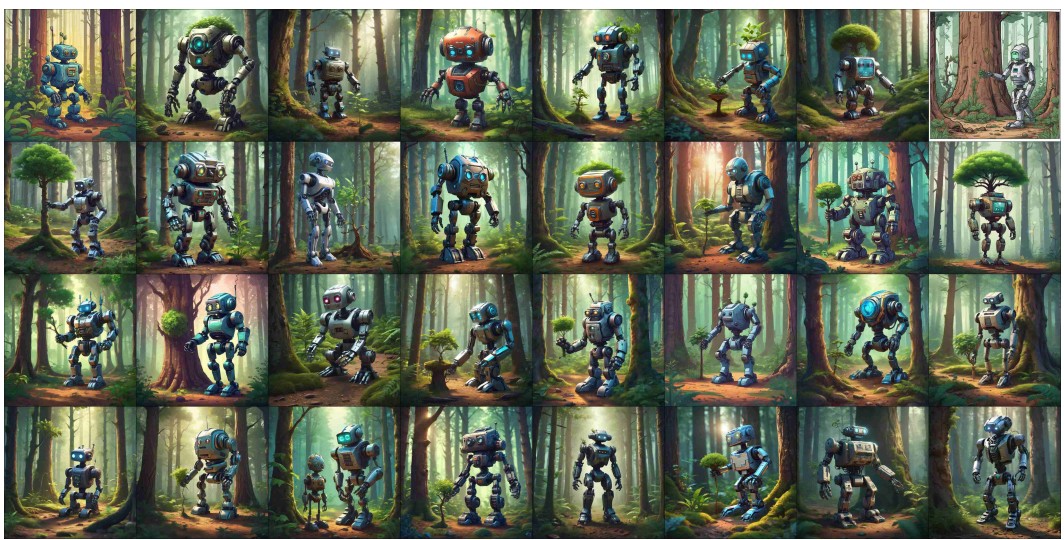

(a) Pre-trained

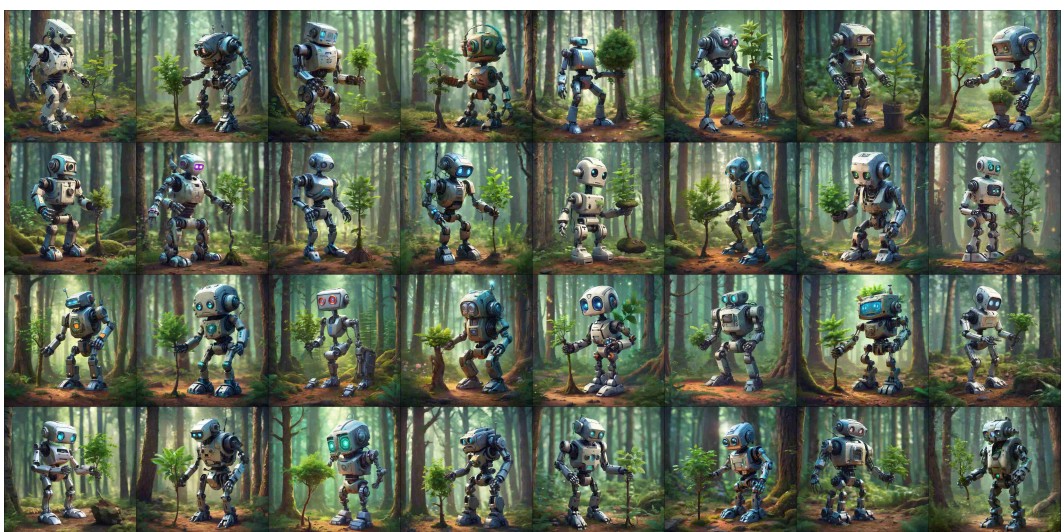

(b) Fast Direct

Figure 28: The 32 randomly generated images for the prompt "robot-plant" guided by Fast Direct by utilizing 50 batch query budget.

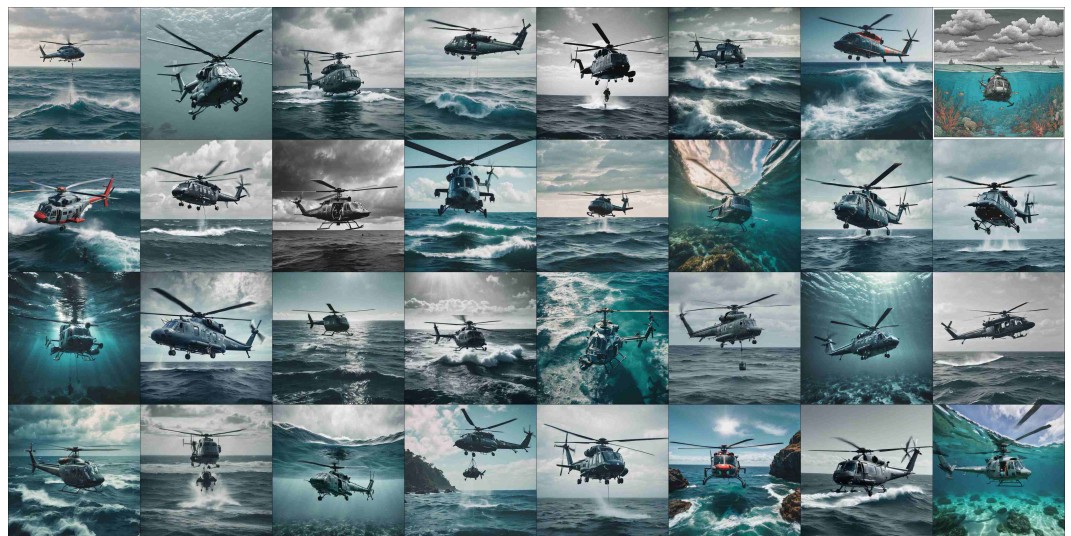

(a) Pre-trained

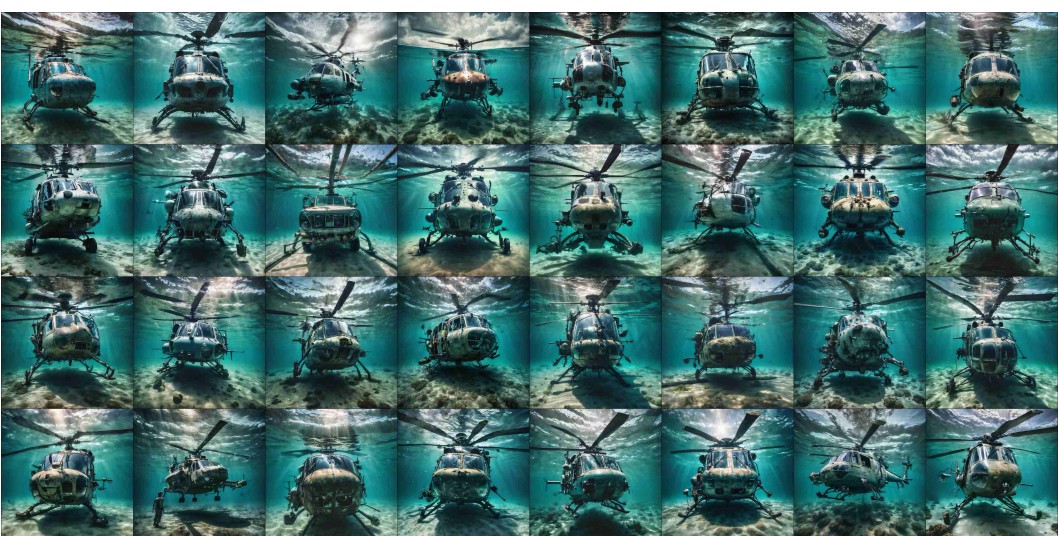

(b) Fast Direct

Figure 29: The 32 randomly generated images for the prompt "ocean" guided by Fast Direct by utilizing 50 batch query budget.

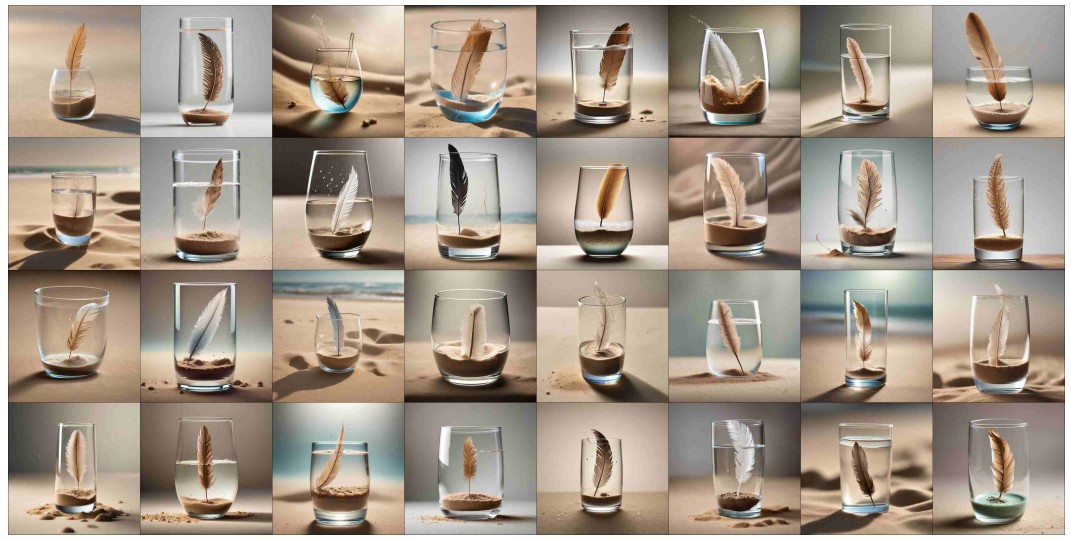

(a) Pre-trained

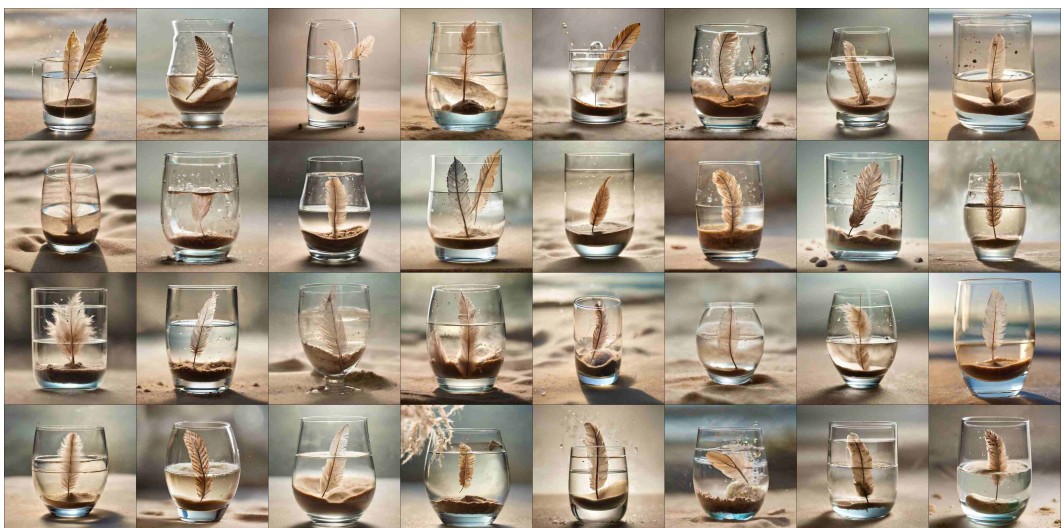

(b) Fast Direct

Figure 30: The 32 randomly generated images for the prompt "sand-glass" guided by Fast Direct by utilizing 50 batch query budget.

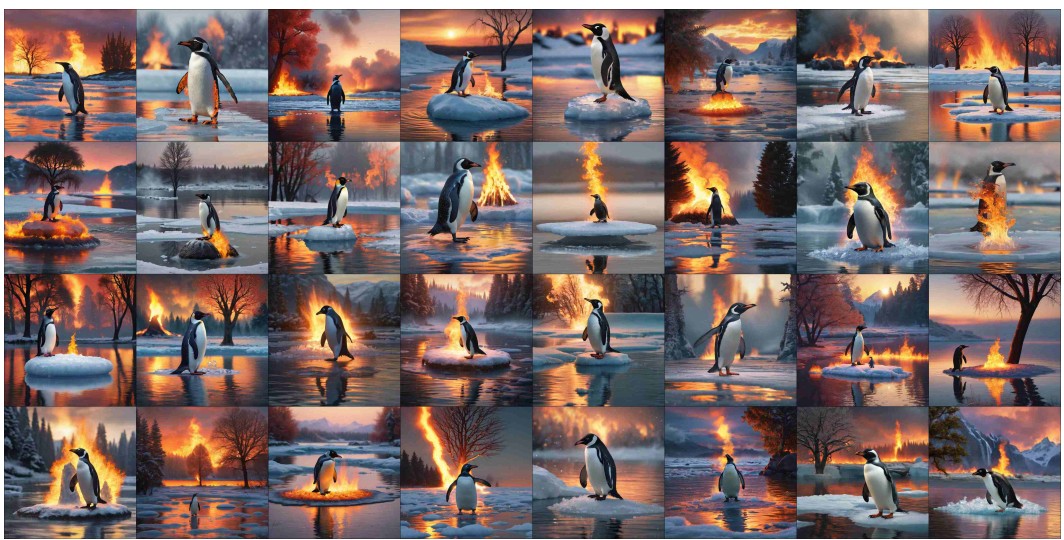

(a) Pre-trained

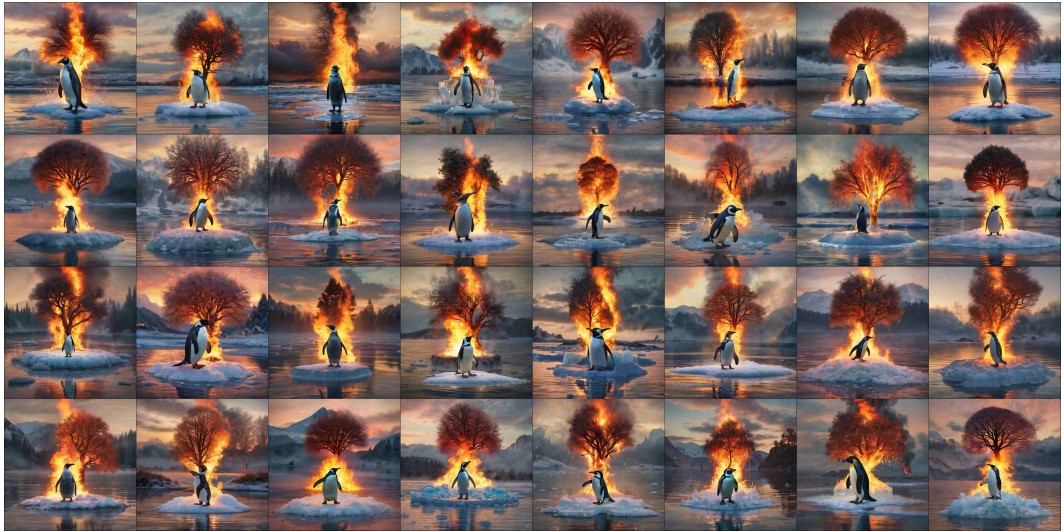

(b) Fast Direct

Figure 31: The 32 randomly generated images for the prompt "penguin" guided by Fast Direct by utilizing 50 batch query budget.

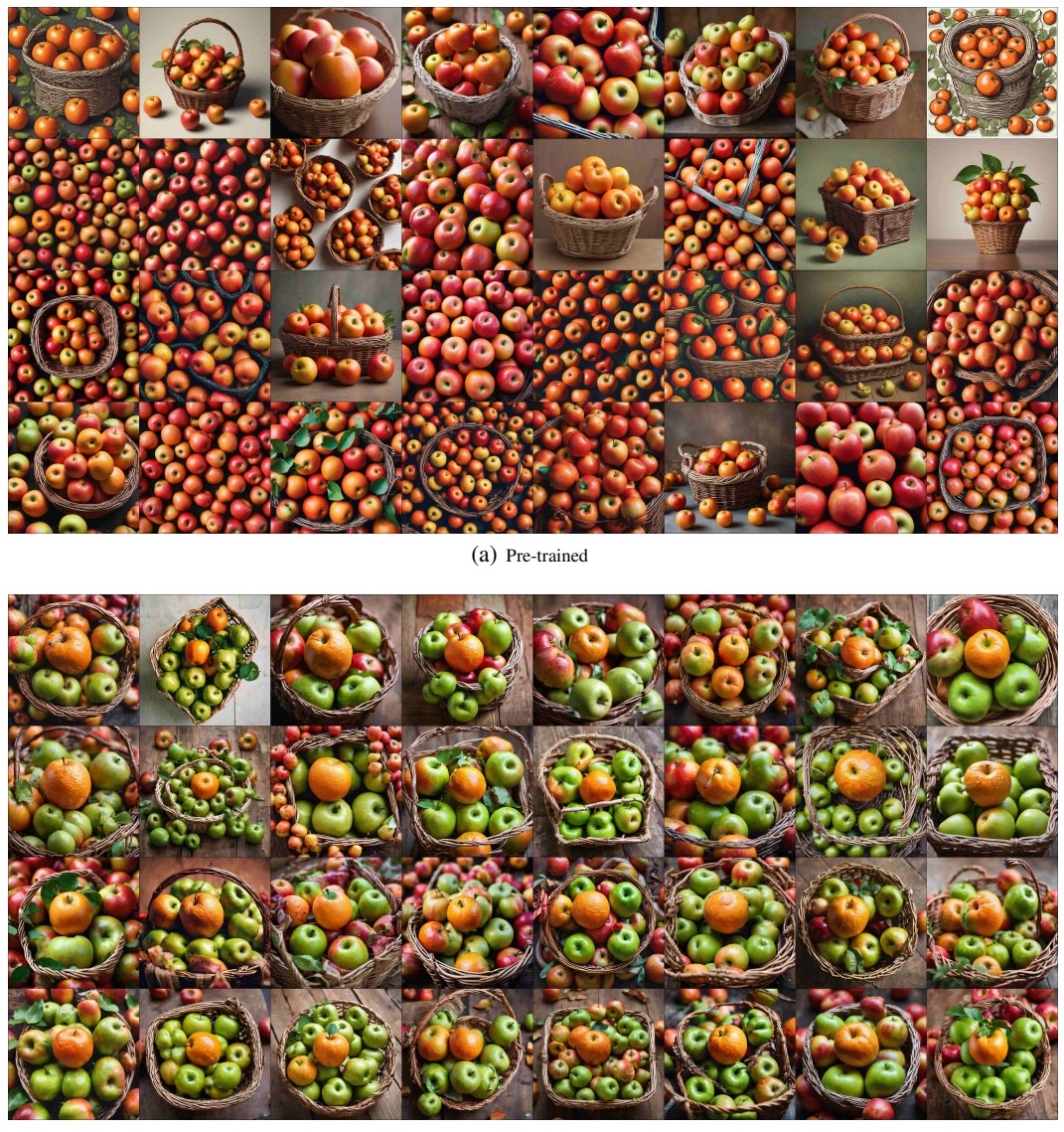

(a) Pre-trained

(b) Fast Direct

Figure 32: The 32 randomly generated images for the prompt "basket" guided by Fast Direct by utilizing 50 batch query budget.

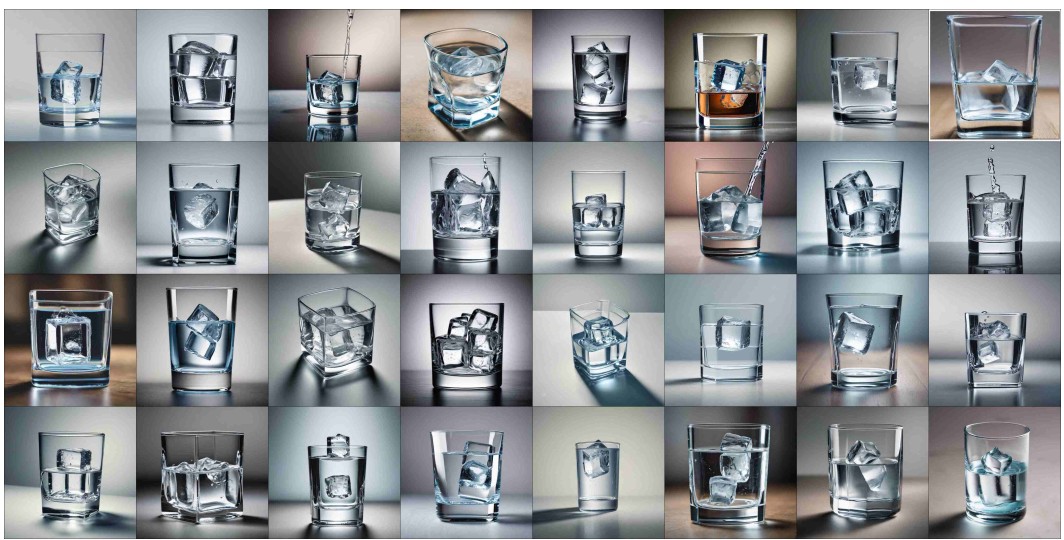

(a) Pre-trained

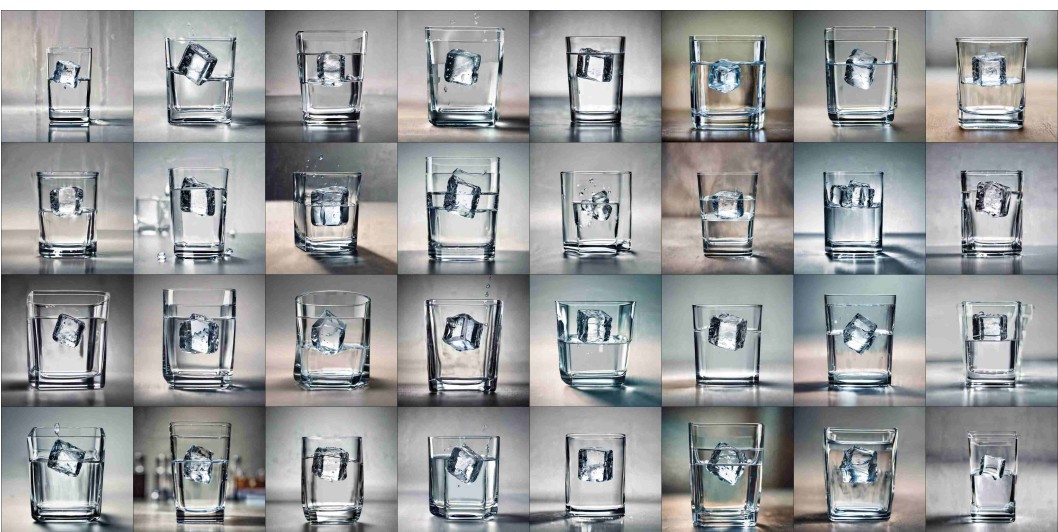

(b) Fast Direct

Figure 33: The 32 randomly generated images for the prompt "ice-cube" guided by Fast Direct by utilizing 50 batch query budget.

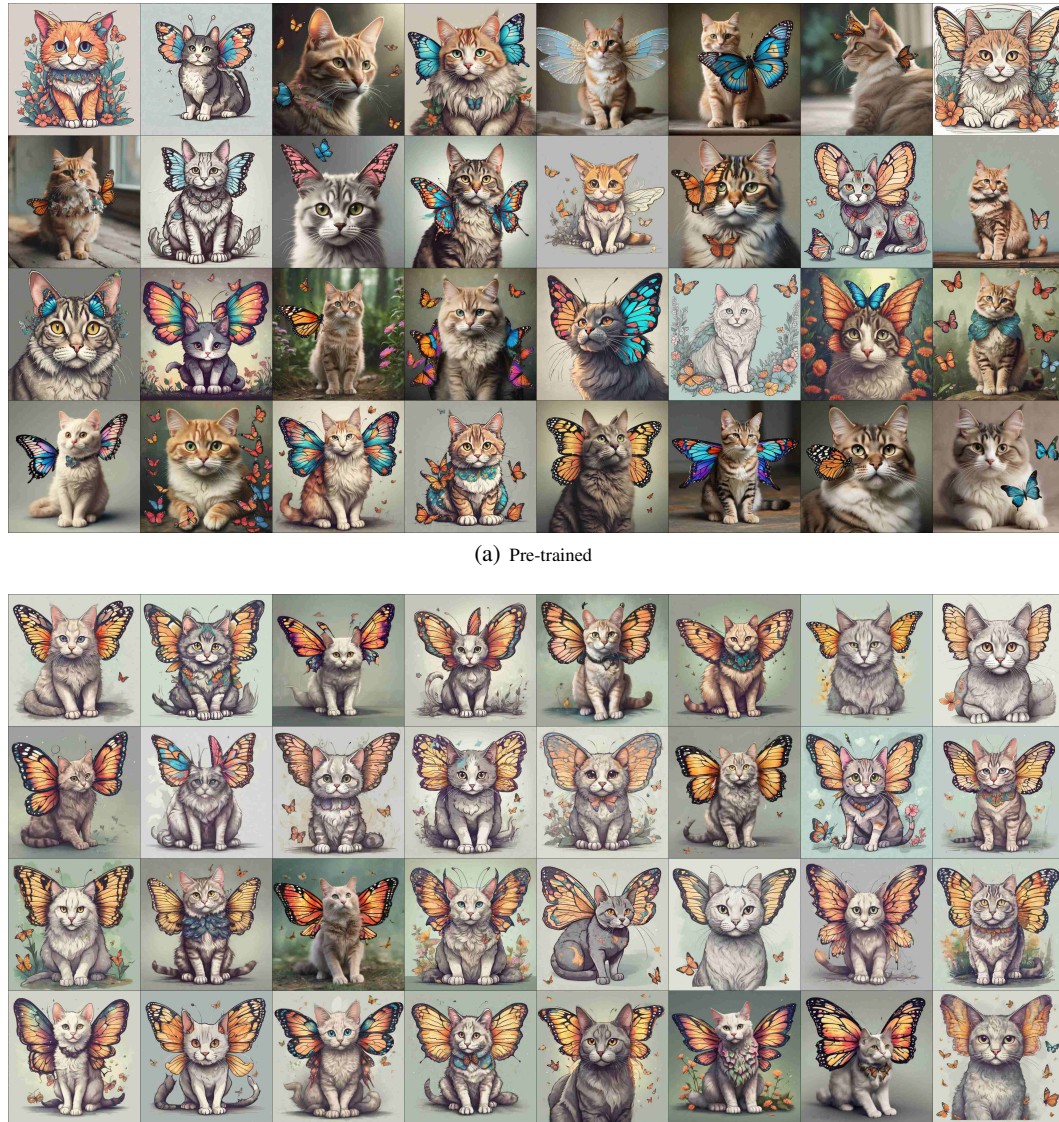

(a) Pre-trained

(b) Fast Direct

Figure 34: The 32 randomly generated images for the prompt "cat-butterfly" guided by Fast Direct by utilizing 50 batch query budget.

