# Supplementary Material

## More Generated Images by Fast Direct for Compressibility, Incompressibility, and Aesthetic Quality

We present the images generated for compressibility, incompressibility, and aesthetic task.

From Fig. 1 to Fig. 16 we present the generated images using the 16 **unseen prompts** by each algorithm in the compressibility task; from Fig. 17 to Fig. 32 we present for incompressibility task; from Fig. 33 to Fig. 47 we present aesthetic task.

From Fig. 48 to Fig. 50 we present 32 randomly generated images by Fast Direct for compressibility, incompressibility, and aesthetic task, respectively.

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

 "cheetah" in aesthetic task, extra batch queries budgets (until 500) are given to the baselines methods for demonstration.

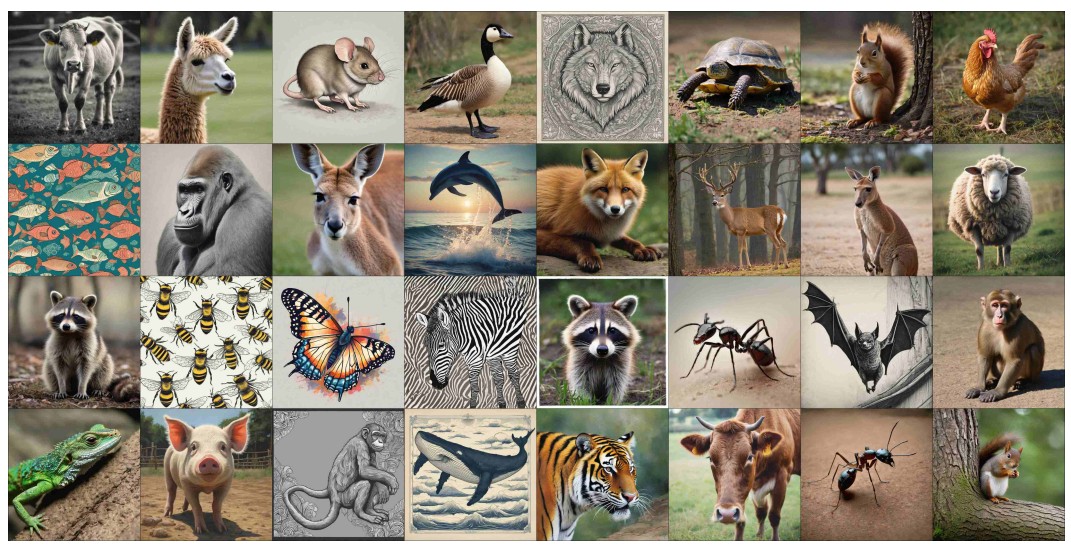

(a) Pre-trained

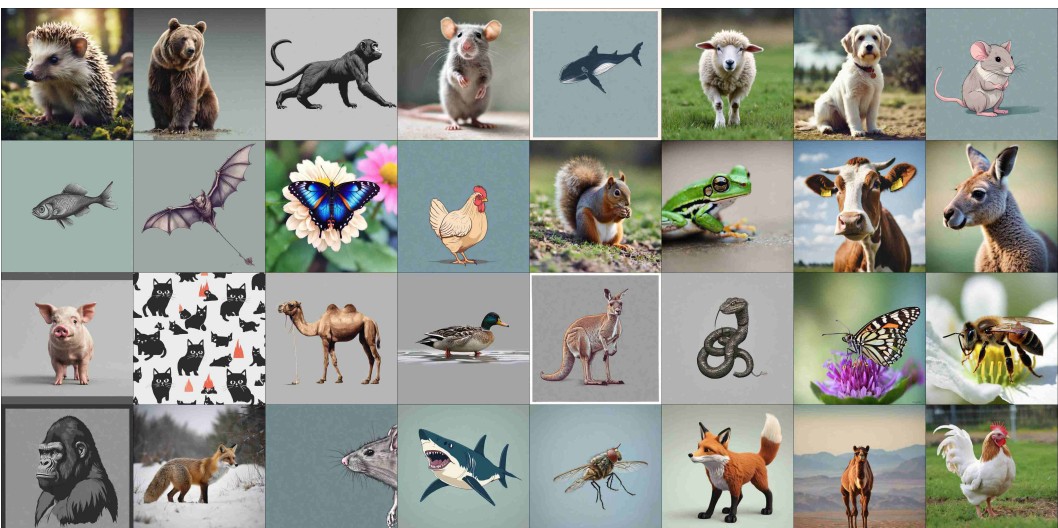

(b) Fast Direct

Figure 48: The 32 randomly generated images for the compressibility task by Fast Direct utilizing 50 batch queries budget.

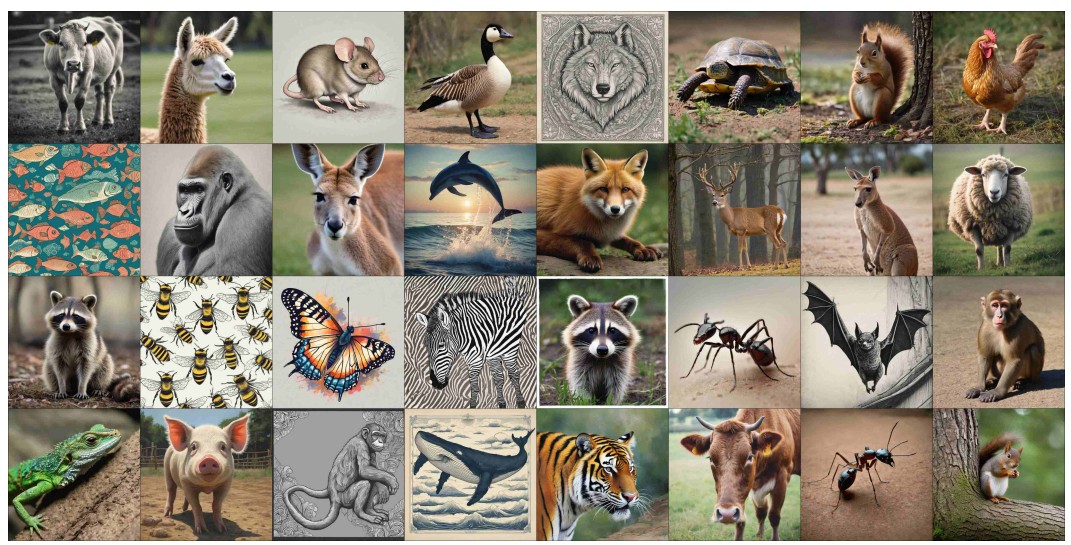

(a) Pre-trained

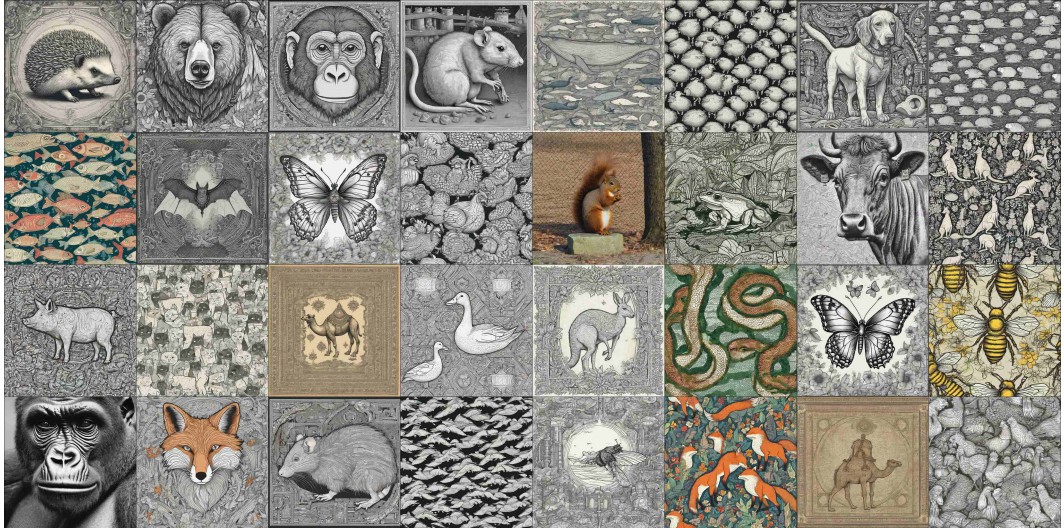

(b) Fast Direct

Figure 49: The 32 randomly generated images for the incompressibility task by Fast Direct utilizing 50 batch queries budget.

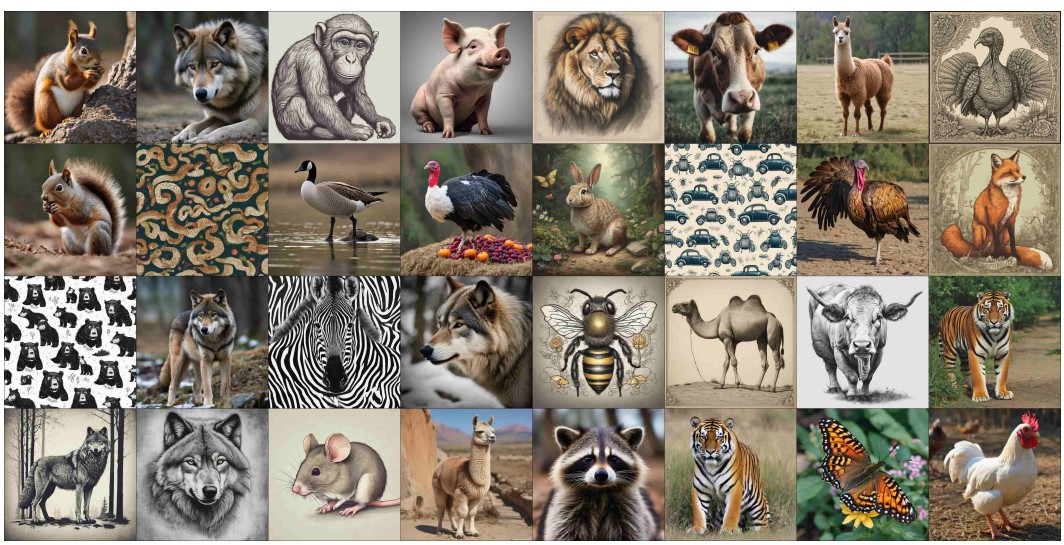

(a) Pre-trained

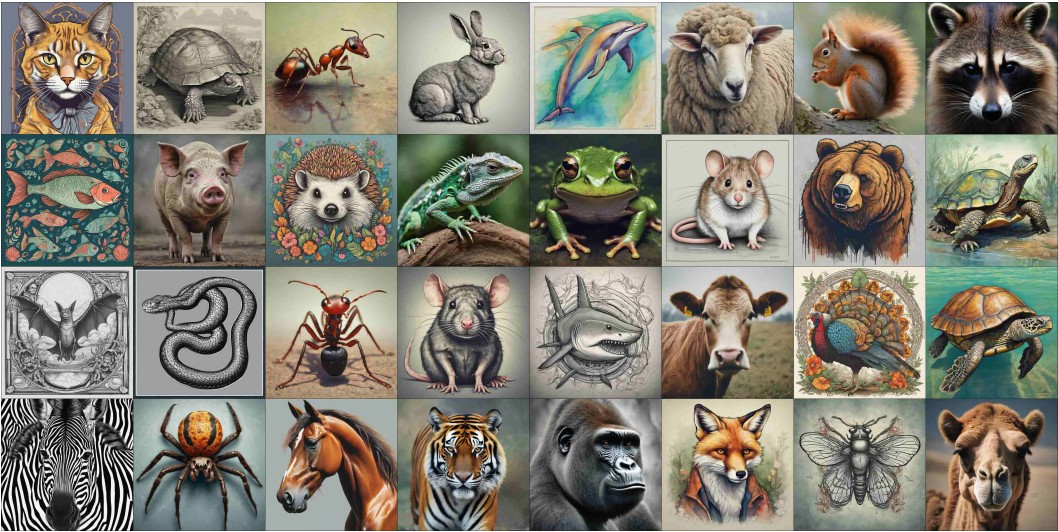

(b) Fast Direct

Figure 50: The 32 randomly generated images for the aesthetic task by Fast Direct utilizing 100 batch queries budget.