# OpenReview forum: "Fast Direct: Query-Efficient  Online Black-box Guidance  for Diffusion-model Target Generation"
_ICLR.cc/2025/Conference — ICLR 2025 Poster_

### Official Review · Reviewer_KTQQ · 2024-10-31

**Soundness:** 3
**Presentation:** 3
**Contribution:** 3
**Rating:** 8
**Confidence:** 3

**Summary:**

This paper introduces an inference-time black-box online guidance method for diffusion models. The authors first propose Guided Noise Sequence Optimization (GNSO), an efficient and robust technique for guiding a diffusion model sampling process toward a desired target sample. Building upon GNSO, they present Fast Direct, a novel approach that leverages pseudo-target samples derived from an online, black-box reward model as guidance targets. This approach enables guidance for any reward model, making it highly versatile. Compared to existing black-box online guidance methods, this approach offers the advantage of inference-time guidance, significantly reducing the number of reward model queries required to steer the sampling process toward the desired target.

**Strengths:**

1. This paper advances the efficiency of reward model queries in steering diffusion models with online black-box reward models. By employing an inference-time guidance method, it avoids the computationally intensive process of model fine-tuning, making it a more practical and efficient solution.
2. From a user perspective, inference-time guidance offers a more immediate and flexible approach to personalization compared to model fine-tuning. This makes the proposed methodology highly promising for real-world applications.
3. The paper proposes a novel approach to online black-box guidance, leveraging reward models to generate pseudo-targets for the GNSO, a technique originally used for same-dimensional target guidance.
4. The proposed methodology has been validated across various domains (image and molecule) and tasks (text-image alignment and drug discovery), demonstrating superior performance and efficiency compared to existing black-box online reward model-based guidance methods.
5. The paper presents its methodology clearly and concisely, making it easy to understand and follow.

**Weaknesses:**

1. The number of experimental tasks is insufficient, and the number of prompts used in the experiment is limited. A direct comparison with reinforcement learning methodologies for diffusion models, such as those in [1], would benefit from evaluating common tasks like compressibility, incompressibility, and aesthetic scores. Additionally, the manual selection of a small number of prompts by the author raises concerns about potential bias toward the proposed methodology.
2. The comparison with fine-tuning baselines may be unfair due to their intended purpose, which differs from the proposed inference-time guidance approach. However, given the absence of other black-box and online inference-time guidance methods, this comparison may be justifiable.
3. The methodology's optimization of noise during sampling is incompatible with the ODE sampler, a popular choice due to its deterministic nature and high performance [2]. This limitation could hinder the method's practical applicability.
4. In the text-image alignment experiments, the proposed method utilized the EDM sampler, while the baseline employed the DDIM sampler. This experimental setup may favor the proposed method, potentially compromising the fairness of the comparison.

[1] Black, Kevin, et al. "Training diffusion models with reinforcement learning." *ICLR* (2024).

[2] Karras, Tero, et al. "Elucidating the design space of diffusion-based generative models." *NeurIPS* (2022).

**Questions:**

1. How was the point in time considered complete for guidance iterations determined? Explicitly stating this criterion within the text would provide greater clarity.
2. Are there any comparison results with the inference time guidance method? Although online black box inference time guidance methods are limited, incorporating a comparison with existing methods, even if the performance of the presented methodology is lower, would offer valuable insights. Additionally, considering the paper's focus on tasks that do not allow backpropagation through the reward model, performing neural network-based reward model tasks (compressibility, incompressibility, and aesthetic scores) commonly used in diffusion model reinforcement learning would make it possible to compare with those methods.
3. What is the number of diffusion sampling steps used in each experiment?
4. What are the specific advantages of guiding $\epsilon_k$ compared to directly guiding $x_k$?
5. The contribution summary in the introduction is not in a clear summary form. It is recommended to reorganize the detailed contributions into a separate paragraph and condense the summary into two or three concise lines.
6. Is the value of $||\epsilon_k||$ used in line 13 fixed as a value pre-calculated in line 3? Or is it a value that continuously changes as $\epsilon_k$ is updated by the loop in line 4?
7. Why was the experiment conducted on $x^{\star} − x_k$ instead of $x^{\star} − \hat{x}_k$ (predicted clean image) in Figure 1 (ablation)? Given that $x^\star$ represents a clean image, comparing it to the predicted clean image $\hat{x}_k$ at the same noise level appears to be a more intuitive approach for assessing the difference.

---

> ### Author Response · Authors · 2024-11-24
>
> We sincerely appreciate the reviewer's constructive advice and valuable comments. Our detailed responses to the reviewer's questions are presented as follows:
>
> **Q1**: The number of experimental tasks is insufficient, and the number of prompts used in the experiment is limited. A direct comparison with reinforcement learning methodologies for diffusion models, such as those in [1], would benefit from evaluating common tasks like compressibility, incompressibility, and aesthetic scores. Additionally, the manual selection of a small number of prompts by the author raises concerns about potential bias toward the proposed methodology.
>
> **A1**: Thanks for the suggestion. We further evaluate compressibility, incompressibility, and aesthetic quality tasks. The experimental results are now included in Appendix A (Page 14). We have used the 45 common animal prompts proposed by [1] to avoid potential experimental bias.
>
> From the experiments, we observe that Fast Direct achieves better scores than the baselines. Moreover, we observe that Fast Direct with the GP model has a generalization ability for unseen prompts. In contrast, DNO is an instance-level method, which cannot generalize to generate images using unseen prompts without re-running the query (optimization) process.
>
> [1] Black, Kevin, et al. "Training diffusion models with reinforcement learning." ICLR(2024).
>
> **Q2**: The methodology's optimization of noise during sampling is incompatible with the ODE sampler, a popular choice due to its deterministic nature and high performance [2]. This limitation could hinder the method's practical applicability.
>
> **A2**: Thank you for raising this point. Fast Direct indeed relies on a stochastic diffusion process for effective stochastic exploration. This requirement is shared by other approaches such as DDPO, DPOK, and D3PO, which also depend on stochastic diffusion for exploration. However, unlike these methods, which require closed-form calculations of the log probability for each diffusion step and therefore being limited to specific samplers like DDIM; Fast Direct treats the SDE sampler as a black-box. This flexibility allows Fast Direct to directly support any SDE sampler.
>
> **Q3**: In the text-image alignment experiments, the proposed method utilized the EDM sampler, while the baseline employed the DDIM sampler. This experimental setup may favor the proposed method, potentially compromising the fairness of the comparison.
>
> **A3**: We further evaluate Fast Direct with the DDIM sampler in image experiments. The experimental results are updated in Fig. 2 and Fig. 4 of the main paper and Appendix A. This would ensure a fair comparison. From the experiment, we observed that Fast Direct achieves similar performance with the DDIM sampler as it does with the EDM sampler.
>
> **Q4**: How was the point in time considered complete for guidance iterations determined? Explicitly stating this criterion within the text would provide greater clarity.
>
> **A4**: Thank you for the question. In our experiment, the guidance iterations stop once the pre-set batch query budget (N) is exhausted. In practice, as it is an online learning algorithm, the user may choose to terminate the guidance integration upon achieving satisfactory results.
>
> **Q5**: Are there any comparison results with the inference time guidance method? Although online black box inference time guidance methods are limited, incorporating a comparison with existing methods, even if the performance of the presented methodology is lower, would offer valuable insights.
>
> **A5**: Thanks for the suggestion. We further include DNO, an inference-time guidance method, as our new baseline method. The experiment results are updated in Fig. 2, Fig. 4 of the main paper, and Appendix A.
>
> It is important to note that DNO runs at the instance level, meaning each run produces only one image, and its performance is highly dependent on the initial prior. To ensure a more accurate evaluation, we run 16 independent experiments to generate 16 images and reported the average results. This evaluation required 16$\times$ more batch queries compared with our Fast Direct and all other baselines.
>
> From the experiments, we observe that Fast Direct achieves better objective values than DNO. Furthermore, the learned GP in Fast Direct generalizes well to unseen prompts, whereas DNO being an instance-level method, cannot generalize to generate images using unseen prompts without re-running the query (optimization) process.
>
> **Q6**: What is the number of diffusion sampling steps used in each experiment?
>
> **A6**: We use K=8 diffusion sampling steps for all image experiments (stated in Section 4.1: Pre-trained Model), and K=200 for molecules experiments (stated in Section 4.2: Pre-trained Model).

---

> ### Author Response · Authors · 2024-11-24
>
> **Q7**: What are the specific advantages of guiding  $\epsilon_k$ compared to directly guiding $x_k$?
>
> **A7**: Guiding on the noise space allows us to project the guided $\epsilon_k$ back to its original scale. This serves as a simple regularization technique to prevent the guided $\epsilon_k$ from deviating too far from its initial value. Moreover, it is convenient as it does not require additional hyper-parameters.
>
> Additionally, since the diffusion model is trained with $\epsilon_k \sim \mathcal{N}(0, I)$, where the norm follows a Chi distribution, $||\epsilon_k|| \sim \mathcal{X}_d $, projecting the guided noise back to its original scale ensures that the guided noise also adheres to the same Chi distribution, $|| {\epsilon’}_k || \sim  \mathcal{X}_d $.  This may help prevent the guided $x_k$ from falling outside the support of the pre-trained model, which could otherwise lead to degeneration.
>
> **Q8**: The contribution summary in the introduction is not in a clear summary form. It is recommended to reorganize the detailed contributions into a separate paragraph and condense the summary into two or three concise lines.
>
> **A8**: Thanks for the suggestion, we have revised the contribution summary accordingly.
>
> **Q9**: Is the value of used in line 13 fixed as a value pre-calculated in line 3? Or is it a value that continuously changes as is updated by the loop in line 4?
>
> **A9**: The value of $|| \epsilon_k ||$ used in line 13 is fixed to the pre-calculated value from line 3.
>
> **Q10**: Why was the experiment conducted on instead of (predicted clean image) in Figure 1 (ablation)? Given that represents a clean image, comparing it to the predicted clean image at the same noise level appears to be a more intuitive approach for assessing the difference.
>
> **A10**: Thanks for the suggestion. We further evaluate the suggested $\hat{x}_k$ case. The results are updated to Figure 1 (page 5). From the experiment, we observe that it can produce target images of comparable quality. This insight inspires an alternative guidance direction within our Fast Direct framework, which has the potential to further enhance performance. We leave this as one of our future work.

---

> > ### Comment · Reviewer_KTQQ · 2024-11-25
> >
> > Thank you for addressing most of my concerns. I still have one question:
> >
> > You mentioned that DNO is an instance-level method, but Fast Direct is not. Could you please explain why Fast Direct is not an instance-level method?
> >
> > If this is clarified, I will seriously consider increasing my final score.

---

> ### Author Response · Authors · 2024-11-25
>
> Thanks for the reviewer's feedback.  We hope the following detailed clarification addresses your question.
>
> **Definition.** We refer **instance-level method** as an approach that only optimizes a single data instance in a complete execution cycle. Although the process can be parallelized, each instance is optimized independently, without sharing information across instances. In contrast, a **batch-level method** optimizes multiple data instances simultaneously in a single execution cycle, often sharing the learned information to reduce redundant efforts across instances.
>
> Here we briefly describe the **DNO** for black-box objective. Let $\mathcal{E} = [\epsilon_0^\top, \cdots, \epsilon_K^\top]$ denote the noise vector instance, where $\epsilon_0$ represents the diffusion prior, and $\lbrace \epsilon_1, \cdots, \epsilon_K \rbrace$ represent the intermediate diffusion noise. The generated image $x$ is uniquely determined by the diffusion model as $x = M_\theta(\mathcal{E})$. The noise vector can be optimized w.r.t. the objective $f(x)$ by gradient descent: $\mathcal{E} \gets \texttt{optimizer-step}(\mathcal{E}, \nabla_{\mathcal{E}} f(x))$. Since $f$ is black-box, DNO uses ZO-SGD to approximate the gradient. Specifically, the gradient w.r.t. $x$ is estimated as:
>
> $$
>     \nabla_x f(x) \approx \hat{H}(x) = \frac{1}{q} \sum_{i=1}^{q} \left[ \left(f(x_i) - f(x)\right) (x_i - x) \right],
> $$
>
> where each $x_i$ is the image generated by perturbed noise vectors:
>
> $$
>     x_i = M_\theta(\mathcal{E} + \mu \xi_i), \text{ for } i \in \lbrace 1, \cdots, q \rbrace,
> $$
>
> where $\xi_i \sim \mathcal{N}(0, I)$ is sampled from a Gaussian distribution, and $\mu > 0$ is a small coefficient to ensure accurate estimation. The gradient w.r.t. the noise vector is estimated by $\nabla_{\mathcal{E}} f(x) \approx \hat{H}(x) \cdot \nabla_{\mathcal{E}} M_\theta(\mathcal{E})$ to optimize the noise vector. As a result, each gradient estimation consumes $q$ objective evaluations.
>
> After $T$ optimization iterations, DNO consumes $T \times q$ objective evaluations and yields an optimized noise vector $\mathcal{E}'$, which generates one optimized image $x' = M_\theta(\mathcal{E}')$. Since DNO produces one image per execution, it is categorized as an instance-level method.
>
> In contrast, **Fast Direct** optimizes a batch of noise vectors in parallel. In each optimization iteration, a batch of $B$ noise vectors $\lbrace \mathcal{E}_1, \cdots, \mathcal{E}_B \rbrace$ is sampled to generate a corresponding batch of images $\lbrace x_1, \cdots, x_B \rbrace$. These images are then evaluated for their objective values, $\lbrace y_1, \cdots, y_B \rbrace$, where $y_i = f(x_i)$. Then, the pseudo-target model is updated with the batch of labeled data $\lbrace (x_1, y_1), \cdots, (x_B, y_B) \rbrace$ to further guide the diffusion process, which shares the information across all the instances. Notice this step consumes $B$ objective evaluations.
>
> After $T$ iterations, Fast Direct consumes $T \times B$ objective evaluations and generates $B$ images. This makes Fast Direct a batch-level method, as it can generate $B$ images simultaneously. Moreover, at the end of the execution, Fast Direct not only produces images but also produces a pseudo-target model (e.g., GP model) that learns from $T \times B$ data points. This pseudo-target model can be used in Fast Direct to generate new target data (even for new prompts) without requiring additional objective evaluations. Specifically, this can be implemented by running Algorithm 2 without Line 17 (query for black-box objective) and Line 18. We emphasize that the produced pseudo-target model of Fast Direct is now used in Line 11 without further updating.
>
> **Summary.** DNO produces one image per run, so it is an instance-level method. While the process can be parallelized, each image is optimized independently, with no information sharing across instances. To generate $B$ images, DNO requires $T \times B \times q$ objective evaluations.
>
> In contrast, Fast Direct can generate a batch of images in parallel with information shared across all the instances, so it is a batch-level method. To generate $B$ images, it only consumes $T \times B$ objective evaluations. Moreover, at the end of the execution, Fast Direct produces a pseudo-target model that learns from $T \times B$ data points. This pseudo-target model can be used in Fast Direct to generate new target data (even for new prompts) without requiring additional objective evaluations.

---

> > ### Comment · Reviewer_KTQQ · 2024-11-26
> >
> > So, you mean that each pipeline stage of Fast Direct (noise initialization, noise optimization, query, diffusion reverse sampling) is parallelized, except for pseudo-target generation, which can share information within a batch, right?
> >
> > Then, I agree that Fast Direct is a batch-level method instead of an instance-level method. My concerns have been addressed, and new insights have been explored during the rebuttal period. I've therefore increased my final score. Thank you for your efforts.
> >
> > I suggest that Algorithm 2 be more explicit about parallelization and aggregation. While it mentions "batch size B," it doesn't clearly indicate which parts are parallelized and which parts share information.
> >
> > By the way, in the experiments in Appendix A, were the input prompts (e.g., "cat," "dog," ...) kept the same within a batch, or were they also randomly sampled?

---

> > > ### Author Response · Authors · 2024-11-26
> > >
> > > Thanks for the reviewer's positive feedback and increasing the score. For the additional question, we provide the answer as below:
> > >
> > > Yes, the sample generation of Fast Direct is parallelized, and the pseudo-target model shares the information within a batch. We have updated Algorithm 2 to explicitly highlight the parallelization (Line 3) and aggregation for pseudo-target model updating (Line 16), making it clearer.
> > >
> > > For compressibility, incompressibility, and aesthetic quality tasks, we follow DDPO to randomly and independently sample the prompt for each instance within a batch. All the baselines employ the same prompts sampling strategy for fine-tuning/optimization.

---

### Official Review · Reviewer_hGsH · 2024-11-02

**Soundness:** 3
**Presentation:** 3
**Contribution:** 3
**Rating:** 6
**Confidence:** 3

**Summary:**

The paper, titled "Fast Direct: Query-Efficient Online Black-Box Guidance for Diffusion-Model Target Generation," introduces an innovative approach to guiding diffusion models in black-box target generation tasks. It addresses scenarios where the objective functions are non-differentiable and where training datasets are unavailable, making it applicable to real-world applications such as drug discovery and human-preference-driven image generation. The authors propose a new method, Fast Direct, leveraging a Guided Noise Sequence Optimization (GNSO) technique. Through extensive experimentation on image and molecular generation tasks, the paper demonstrates that Fast Direct outperforms several baseline methods in query efficiency and generation quality.

**Strengths:**

Novelty and Relevance: The Fast Direct algorithm introduces a unique approach to guiding diffusion models without requiring differentiable objectives or offline data, making it relevant for applications with black-box or non-differentiable feedback.
Efficiency: The method shows impressive improvements in query efficiency across tasks, which is valuable in settings with expensive evaluations, such as in drug design.
Extensive Experiments: The paper includes rigorous experiments across various image and molecular generation tasks, showing consistent improvements over baseline methods in both query efficiency and task performance.
Generalizability: The proposed approach is designed to be flexible, supporting various black-box objectives and stochastic diffusion schedulers, which suggests it could be adapted to other domains and applications beyond those tested.

**Weaknesses:**

Limited Theoretical Justification: While empirical results are promising, the paper lacks a thorough theoretical analysis of the robustness and convergence properties of Fast Direct’s update strategy. Providing an in-depth theoretical foundation could strengthen the paper’s contributions and clarify the algorithm's potential limitations.
Baseline Comparisons: Although Fast Direct shows efficiency gains over DDPO, DPOK, and D3PO, a discussion is missing on why these methods were specifically chosen as baselines and how representative they are of the current state of the art in black-box guidance.
Scalability and Computational Costs: The paper could provide more detail on the computational overhead introduced by Fast Direct, especially for high-dimensional tasks like image generation. Scalability analysis in terms of computational cost and time efficiency would be beneficial, particularly as query budgets increase.
Lack of Ablation Studies: The paper does not explore the sensitivity of Fast Direct’s performance to its key hyperparameters, such as step size and batch size. Given the flexibility mentioned for the pseudo-target update, ablation studies could reveal the robustness of these parameters and how they influence the results across tasks.

**Questions:**

Why were DDPO, DPOK, and D3PO chosen as baselines? Are they the most relevant and competitive in black-box target generation? If there are other methods that could be appropriate, a brief comparison or justification for their exclusion could enhance the robustness of the evaluation.
Have you considered alternative pseudo-target selection strategies? While the paper presents GP and historical-optimal updates, discussing the potential effects of alternative methods (e.g., reinforcement learning-based or clustering-based approaches) might highlight the generalizability of Fast Direct.
Could you provide more insights into the parameter tuning process for Fast Direct? As query efficiency is a primary focus, understanding the sensitivity of Fast Direct to parameters like batch size and step size would clarify the required fine-tuning efforts.
Could you expand on the potential for scalability? Especially in high-dimensional tasks like high-resolution image generation, it would be helpful to know if there are constraints or limitations for Fast Direct in terms of computational efficiency.

---

> ### Author Response · Authors · 2024-11-24
>
> We sincerely appreciate the reviewer's constructive advice and valuable comments. Detailed responses to the reviewer's questions are given below.
>
> **Q1**: Why were DDPO, DPOK, and D3PO chosen as baselines? Are they the most relevant and competitive in black-box target generation? If there are other methods that could be appropriate, a brief comparison or justification for their exclusion could enhance the robustness of the evaluation.
>
> **A1**: DDPO and DPOK are two representative RL-based methods for fine-tuning diffusion models. Both approaches support online learning and black-box objectives. D3PO focuses on relative human rewards and also supports online learning and black-box objectives. We selected these methods as baselines because they have been shown to effectively handle black-box objectives in an online manner.
>
> Most recently, DNO[1] introduced a method to guide the diffusion process using ZO-SGD, enabling support for black-box objectives. We further include DNO as a new baseline in our experiments. This would enhance the robustness of the evaluation. The new experiments are included in the  Fig. 2 and Fig. 4 of the main paper, and Appendix A.
>
> It is important to note that DNO runs at the instance level, meaning each run produces only one image, and its performance is highly dependent on the initial prior. To ensure a more accurate evaluation, we run 16 independent experiments to generate 16 images and reported the average results. This evaluation required 16$\times$ more batch queries compared with our Fast Direct and all other baselines.
>
> From the experiments, we observe that Fast Direct achieves significantly better objective values than DNO. Furthermore, the learned GP in Fast Direct generalizes well to unseen prompts, whereas DNO being an instance-level method, cannot generalize to generate images using unseen prompts without re-running the query (optimization) process.
>
> [1]: Tang, Zhiwei, et al. "Tuning-Free Alignment of Diffusion Models with Direct Noise Optimization." arXiv preprint arXiv:2405.18881 (2024).
>
> **Q2**: Have you considered alternative pseudo-target selection strategies? While the paper presents GP and historical-optimal updates, discussing the potential effects of alternative methods (e.g., reinforcement learning-based or clustering-based approaches) might highlight the generalization of Fast Direct.
>
> **A2**: Thank you for the suggestion. It is interesting to consider alternative pseudo-target selection strategies under our Fast Direct framework, which may further improve the query efficiency and we believe this would be a promising direction. We leave this as one of our future work.
>
> **Q3**: Could you provide more insights into the parameter tuning process for Fast Direct? As query efficiency is a primary focus, understanding the sensitivity of Fast Direct to parameters like batch size and step size would clarify the required fine-tuning efforts.
>
> **A3**: Thank you for your question. We have conducted an ablation study about the hyper-parameters in Appendix B (page 18), which shows that Fast Direct is not sensitive to the batch size and step size settings. The batch size can be adjusted based on available GPU memory and the query budget.
>
> **Q4**: Could you expand on the potential for scalability? Especially in high-dimensional tasks like high-resolution image generation, it would be helpful to know if there are constraints or limitations for Fast Direct in terms of computational efficiency.
>
> **A4**: Thank you for your question. We have conducted an ablation study on computational time in Appendix B (page 18). The results show that Fast Direct requires approximately 6.4 hours per experiment with a budget of 50 batch queries to generate 32 images simultaneously. Experiments show that Fast Direct scales linearly with batch size for small historical query dataset (e.g., 3.2k data points). However, because Fast Direct relies on a GP model, its computational time scales cubically. This limitation can potentially be addressed using kernel approximation techniques [2,3]. We leave this as one of our future work.
>
> [2] Rahimi  et al. Random Features for Large-Scale Kernel Machines. NeurIPS (NIPS) 2007.
>
> [3] Yu et al. Orthogonal random features. NeurIPS 2016.

---

> ### Author Response · Authors · 2024-11-27
>
> Dear Reviewer hGsH
>
> We truly appreciate the effort and time you have devoted to providing constructive reviews, as well as your positive evaluation of our submission.
>
> We have now provided more clarifications, explanations, and experiments to address your concerns and followed your advice to improve our paper.
>
> Would you mind checking our responses and confirming if they have addressed your concerns? We truly appreciate this opportunity to improve our work and shall be most grateful for any feedback you could give us.

---

### Official Review · Reviewer_dSoP · 2024-11-08

**Soundness:** 3
**Presentation:** 4
**Contribution:** 4
**Rating:** 8
**Confidence:** 3

**Summary:**

This work proposed a new online black-box algorithm, Fast Direct, to provide guidance for diffusion model without the need of pretrained guidance model or differentiable objective function. The important findings of the noise update direction in Guided Noise Sequence Optimization laid the foundation of the Fast Direct algorithm. The Fast Direct showed consistent query efficiency compared to 3 baselines.

**Strengths:**

1. This work used a data driven method for image generations, different from the three baselines in previous works focusing more on finetuning the diffusion model. This should be a good indicator of originality.

2. The proposed solution was a training free solution, which also led to a more query efficient method.

**Weaknesses:**

In case of the query being OOD, and the Gemini continues to dislike the generated image, the proposed solution might be less efficient compared to the methods involved finetuning the diffusion model.

**Questions:**

1. For comparison with three baselines, were there reward models finetuned with Gemini scores? Or were the Gimini scores directly used for finetuning the diffusion model? It was not really clear to me from my reading.

2. In the claim of selecting K instead of k for the update direction in Section 3.1 and the observation from Figure 1, I'd really love to also see the effect of choosing K/2 and K/4 to get a sense how the degree of "far away from the data manifold" correlates quality degradation.

3. While aiming to deal with the situation of missing a guidance model, could you also add a comparison of the results using this method versus using a trained guidance model? That would make the story more convincing.

---

> ### Author Response · Authors · 2024-11-24
>
> We sincerely appreciate the reviewer's constructive advice and valuable comments. Your support is greatly appreciated. Detailed responses to the reviewer's questions are given below.
>
> **Q1**: For comparison with three baselines, were there reward models finetuned with Gemini scores? Or were the Gimini scores directly used for finetuning the diffusion model? It was not really clear to me from my reading.
>
> **A1**: We have clarified the baselines details in Appendix D (page 18). All baselines use the Gemini rating to directly fine-tune the diffusion model. No additional reward models are trained/fine-tuned for all methods.
>
> **Q2**: In the claim of selecting K instead of k for the update direction in Section 3.1 and the observation from Figure 1, I'd really love to also see the effect of choosing K/2 and K/4 to get a sense how the degree of "far away from the data manifold" correlates quality degradation.
>
> **A2**: Thank you for the suggestion. We have included this experiment in Appendix C (page 18).  We observe that the generated image quality decreases as $K’$ decreases. This is because as $K’$ decreases, the images $\boldsymbol{x}_{K'}$ become noisier, and move further away from the data manifold.
>
> **Q3**: While aiming to deal with the situation of missing a guidance model, could you also add a comparison of the results using this method versus using a trained guidance model? That would make the story more convincing.
>
> **A3**: Thanks for the suggestion. Due to a lack of data before the query process, we were unable to obtain a pre-trained guidance model. Instead, we have added DNO [1], an online guidance-based approach, as a new baseline method for image tasks. The new experiment results are shown in Fig. 2 and Fig. 4 of the main paper, and Appendix A.
>
> It is important to note that DNO runs at the instance level, meaning each run produces only one image, and its performance is highly dependent on the initial prior. To ensure a more accurate evaluation, we performed 16 independent experiments to generate 16 images and reported the average results. This evaluation required 16$\times$ more batch queries compared with our Fast Direct and all other baselines.
>
> From the experiments, we observe that Fast Direct achieves significantly better objective values than DNO. Furthermore, the learned GP in Fast Direct generalizes well to unseen prompts, whereas DNO, being an instance-level method, cannot generalize to generate images using unseen prompts without re-running the query (optimization) process.
>
> [1]: Tang, Zhiwei, et al. "Tuning-Free Alignment of Diffusion Models with Direct Noise Optimization." arXiv preprint arXiv:2405.18881(2024).

---

> > ### Comment · Reviewer_dSoP · 2024-11-25
> >
> > Thank you for addressing the comments and requests.
> >
> > Please make sure to point Appendix D in Chapter 4 in the main context.

---

> > > ### Author Response · Authors · 2024-11-25
> > >
> > > Apologies for missing that, we have revised accordingly. Thank you for the feedback.

---

### Official Review · Reviewer_TjGV · 2024-11-11

**Soundness:** 2
**Presentation:** 3
**Contribution:** 2
**Rating:** 5
**Confidence:** 1

**Summary:**

This paper proposed an LLM-scored/non-differentiable-scored pipeline for generating aligned outputs via diffusion models during the inference procedure. The paper utilizes Gaussian Process (GP) estimation or historical optimal to insert the guidance of the score. With two different generative tasks, the method outperforms several inference-time diffusion guidance methods with fewer iteration steps.

**Strengths:**

The method utilizes GP/historical optimal as the estimation of the guidance direction, illustrating clear improvements based on the score evaluation. The Bayesian-based optimization in the field of inference-time diffusion guidance is shown to be a useful tool for improving generation quality.

**Weaknesses:**

Lack of ablation studies of the hyper-parameters used in the proposed pipeline (step-size alpha, lambda). Additional experiments on the chosen hyper-parameters (especially for the step-size alpha since this is the guidance scale) are encouraged.

For both image and molecule generation tasks, the evaluation is based on the same metric used in the optimization, which may enrol incomplete or unfair comparisons to the other baselines. The paper could use the same LLM score during the inferences of the baselines (if it is available for them) or post additional evaluation metrics (such as the score from different LLMs and human evaluation scores) to strengthen the argument.

**Questions:**

In Figure 1 (also in the figures in the Appendix), the initial image of the proposed method is different to the other baselines, which seems to be a factor that influences the final generation performance. Can the authors clarify here?
(comment: several references could be listed here).

---

> ### Author Response · Authors · 2024-11-24
>
> We thank the reviewer for dedicating time and effort to review our paper. Detailed responses to the reviewer's questions are given below.
>
> **Q1**: Lack of ablation studies of the hyper-parameters used in the proposed pipeline (step-size alpha, lambda). Additional experiments on the chosen hyper-parameters (especially for the step-size alpha since this is the guidance scale) are encouraged.
>
> **A1**: Thank you for the suggestion. We further include a detailed ablation study about the step size and batch size in Appendix B (Page 18). From the experiment, we observe that Fast Direct is not sensitive to the step size and batch size settings.
>
> **Q2**: For both image and molecule generation tasks, the evaluation is based on the same metric used in the optimization, which may enrol incomplete or unfair comparisons to the other baselines. The paper could use the same LLM score during the inferences of the baselines (if it is available for them) or post additional evaluation metrics (such as the score from different LLMs and human evaluation scores) to strengthen the argument.
>
> **A2**: We clarify that all baselines are optimized and evaluated by the same metric. For image tasks, we use the same LLM score, Gemini rating, to evaluate the generated images; for molecule tasks, we use the Vina docking score to evaluate the generated molecules. The evaluation is consistent across all baselines and Fast Direct to ensure a fair comparison.
>
> **Q3**: In Figure 1 (also in the figures in the Appendix), the initial image of the proposed method is different to the other baselines, which seems to be a factor that influences the final generation performance. Can the authors clarify here?
>
> **A3**: The initial image is slightly different because we use EDM sampler for Fast Direct but use DDIM sampler for baselines. We have added DDIM sampler for Fast Direct in image experiments to ensure the initial images are consistent with the baselines. The results are now included in Fig. 2 and Fig. 4 of the main paper, as well as in Appendix A. This would ensure a fair comparison. From the experiment, we observed that Fast Direct achieves similar performance with the DDIM sampler as it does with the EDM sampler.

---

> ### Author Response · Authors · 2024-11-27
>
> Dear Reviewer TjGV
>
> Thank you again for your time and efforts in reviewing our submission!
>
> We have now provided more clarifications, explanations, experiments, and discussions to address your concerns and followed the advice of all reviewers to improve our paper.
>
> Would you mind checking our responses and confirming if they have addressed your concerns?

---

### Author Response · Authors · 2024-11-24

We thank all reviewers for their valuable comments and appreciate the great efforts made by all reviewers, ACs, SACs and PCs.

In the following, we provide a summary of our updates and responses. For details, please refer to the feedback of each comment/question point-by-point.

- **Suggestion for compressibility, incompressibility, and aesthetic quality tasks:** We further evaluate all these suggested tasks and present experimental results in Appendix A (Page 14) of the updated paper.

- **Concern about the lack of an ablation study:** In response, we further include a detailed ablation study in Appendix B (Page 18) of the updated paper.

- **Suggestion to compare with guidance-based methods:** We further include  DNO (Tang et al. 2024), a recent online guidance-based approach supporting black-box objectives, as a new baseline for the image experiments. These comparisons are now included in Fig. 2, Fig. 4 in the main paper and Appendix A.

- **Concern about unfair comparisons due to diffusion sampler discrepancy:** To ensure fairness, we further include experiments of Fast Direct using the DDIM sampler in image tasks. The experimental results are updated in Fig. 2, Fig. 4 in the main paper and Appendix A.

---

### Meta-Review · Area_Chair_7yXu · 2024-12-22

**Metareview:**

This paper proposes Fast Direct, a novel online black-box algorithm for guiding diffusion models in target generation tasks with non-differentiable objectives and without requiring pretrained guidance models or training datasets. The method builds on the Guided Noise Sequence Optimization (GNSO) technique, using Gaussian Process estimation or historical optima to incorporate guidance during inference. Extensive experiments on image and molecular generation tasks demonstrate its superior query efficiency and generation quality compared to several baseline methods, while requiring fewer iteration steps.

Strength: The proposed Fast Direct algorithm is interesting and innovative, which provides an efficient, training-free, inference-time guidance solution for diffusion models in black-box scenarios with non-differentiable objectives. By leveraging Guided Noise Sequence Optimization (GNSO) and Bayesian-based optimization methods, it achieves improved generation quality and query efficiency. The approach can be flexible, enabling guidance from various reward models without requiring offline data or model fine-tuning, making it highly practical for diverse applications like drug discovery. Extensive experiments across image and molecular generation tasks validate its superior performance and efficiency compared to existing baseline methods.

Weakness: Reviewers raise several weaknesses about the original version of the paper, including the absence of ablation studies to analyze the sensitivity of key hyperparameters, such as step size and batch size, not sufficient details about the computational overhead or scalability of Fast Direct, comparison with fine-tuning baselines and diffusion sampler discrepancy. It also lacks a thorough theoretical justification of the robustness and convergence properties of the Fast Direct algorithm, limiting the understanding of its foundational strengths and potential limitations.

Overall, the proposed method in the paper is quite interesting to provide test-time guidance for the diffusion model generation, demonstrating efficiency in different generation tasks. Most of the concerns raised by reviewers have been well addressed in the rebuttal and revised manuscript. However, I would support that a thorough theoretical analysis would be appreciated to show the robustness and convergence properties of Fast Direct’s update strategy, while this comment seems to be neglected by the author in the rebuttal. But the paper still stands out as an interesting work. Thus, I would lean to accept.

**Additional Comments On Reviewer Discussion:**

Most reviewers have responded to the author's rebuttal and involve in the discussion actively. Most of the concerns raised by reviewers have been well addressed in the rebuttal and revised manuscript. However, I would support that a thorough theoretical analysis would be appreciated to show the robustness and convergence properties of Fast Direct’s update strategy, while this comment seems to be neglected by the author in the rebuttal. But the paper still stands out as an interesting work. Thus, I would lean to accept.

---

### Decision · Program_Chairs · 2025-01-22

Accept (Poster)